# Genetic variation influences food-sharing sociability in honey bees

Ian M. Traniello[1¤*], Arian Avalos[2], Michael J. M. Gachomba[3], Tim Gernat[1], Zhenqing Chen[1], Amy C. Cash-Ahmed[1], Adam R. Hamilton[1], Jennifer L. Cook[3], Gene E. Robinson[1,4,5*]

**1** Carl R Woese Institute for Genomic Biology, University of Illinois at Urbana-Champaign, Urbana, Illinois, United States of America, **2** United States of America Department of Agriculture – Agricultural Research Services, Honey Bee Breeding, Genetics and Physiology Research Laboratory, Baton Rouge, Louisiana, United States of America, **3** School of Psychology, University of Birmingham, Birmingham, United Kingdom, **4** Neuroscience Program, University of Illinois at Urbana-Champaign, Urbana, Illinois, United States of America, **5** Department of Entomology, University of Illinois at Urbana-Champaign, Urbana, Illinois, United States of America

¤ Current Address: Lewis-Sigler Institute for Integrative Genomics, Princeton University, Princeton, New Jersey, USA
* it4770@princeton.edu (IMT); generobi@illinois.edu (GER)

## Abstract

Individual variation in sociability is a central feature of every society. This includes honey bees, with some individuals well connected and sociable, and others at the periphery of their colony's social network. However, the genetic and molecular bases of sociability are poorly understood. Trophallaxis—a behavior involving sharing liquid with nutritional and signaling properties—comprises a social interaction and a proxy for sociability in honey bee colonies: more sociable bees engage in more trophallaxis. Here, we identify genetic and molecular mechanisms of trophallaxis-based sociability by combining genome sequencing, brain transcriptomics, and automated behavioral tracking. A genome-wide association study (GWAS) identified 18 single nucleotide polymorphisms (SNPs) associated with variation in sociability. Several SNPs were localized to genes previously associated with sociability in other species, including in the context of human autism, suggesting shared molecular mechanisms of sociability. Variation in sociability also was linked to differential brain gene expression, particularly genes associated with neural signaling and development. Using comparative genomic and transcriptomic approaches, we also detected evidence for divergent mechanisms underpinning sociability across species, including those related to reward sensitivity and encounter probability. These results highlight both potential evolutionary conservation of the molecular roots of sociability and points of divergence.

**Data availability statement:** Raw and processed genomic and transcriptomic data are publicly available on NCBI under Bioproject PRJNA1237396.

**Funding:** This work was supported by the European Union's Horizon 2020 Research and Innovation Program under ERC-2017-StG Grant Agreement 757583 (Brain2Bee; to JLC and GER) and an Agriculture Research Service Award (8042-21000-291-047S, to GER). IMT is presently supported by the Lewis-Sigler Institute for Integrative Genomics as a Lewis-Sigler Scholar. The funders had no role in study design, data collection and analysis, decision to publish, or preparation of the manuscript.

**Competing interests:** The authors have declared that no competing interests exist.

**Abbreviations:** CNN, convolutional neural network; DEGs, differentially expressed genes; DWV, Deformed Wing Virus; GRM, genetic relatedness matrix; GWAS, genome-wide association study; HCR, Hybridization Chain Reaction; HPGs, hypopharyngeal glands; iPDC, information partial directed coherence; KC, Kenyon cells; LD, Linkage disequilibrium; MB, mushroom body; MRJP, major royal jelly protein; PBS, phosphate-buffered saline; PRS, polygenic risk score; RH, relative humidity; SFARI, Simons Foundation Autism Research Initiative; SNPs, single nucleotide polymorphisms; VAR, vector autoregressive model.

## Introduction

Consistent variation among individuals in the tendency to engage one another is a central feature of all animal societies. "Sociability," which broadly characterizes associative interactions outside of the context of aggression or courtship [1], falls on a spectrum, with hypo- and hyper-sociable individuals flanking a "normal" intermediate range [2]. There are many causes of this variation, stemming from a combination of labile drivers like motivational state [3], social status [4], and previous experience [5,6], as well as heritable factors that drive lifelong positive or negative inclinations toward social interactions [7].

As a trait, sociability arises from the integration of socially responsive neural circuitry, physiology, and experience, all of which contribute to how an animal interprets and responds to social cues [8–10]. Detailed study of vertebrate brain evolution has outlined a "social behavior network" that regulates multiple forms of social behavior [11], acting in concert with the mesolimbic reward system to influence how an animal makes decisions in the context of various social situations [2,12,13].

Social insects, including the western honey bee (*Apis mellifera*), have tiny and very differently organized nervous systems relative to vertebrates, but they also form large and intricate societies whose ecological and evolutionary success are tightly linked to individual- and group-level sociability [14,15]. Honey bees form large colonies of tens of thousands of individuals and are a powerful model for further understanding the relationship between genes and sociability. The bulk of the colony is composed of sterile workers who specialize in different tasks at different ages. Task performance is influenced by a combination of individual age, genetics, and the needs of the colony [16–18]. Task specialization is also related to specific physiological states, with highly conserved signaling molecules like juvenile hormone and vitellogenin acting as modulators of behavioral maturation among worker bees [19]. Changes in lipid titers and the volume of food-producing exocrine glands also contribute to a complex array of physiological mechanisms that shape worker behavior [20].

Honey bee workers lack morphologically distinct castes, and each individual maintains the ability to perform all tasks required for colony growth and development other than personal reproduction. The molecular mechanisms underlying specific behavioral states like brood care ("nursing"), guarding the hive entrance, or foraging outside for floral resources are sensitive to environmental demands, with clear individual differences in the likelihood of performing certain tasks as well as predisposition toward social contact [5]. Honey bee colony organization is largely based on an age-related division of labor: worker bees live 4–6 weeks, performing in-hive tasks like brood care and nest maintenance over the first 1–2 weeks of life before transitioning to out-of-hive tasks like nest defense and foraging for the remainder of their lives [21]. The transition through these behavioral states is shaped by developmental programming and environmental influence, and division of labor emerges even among age-matched cohorts, meaning task distribution is responsive to the needs of the colony [22]. We used honey bees to study genetic and molecular components of sociability, including comparative genomic analyses.

Comparative genomics offers a toolkit for exploring the extent to which analogous behaviors, like the tendency to engage socially with others, have similar molecular representations in the brain across disparate species [23,24]. While a behavioral trait may be similarly expressed across species, whether that trait also has a common molecular basis requires observation of similarities across deeper layers of biological organization, including at the level of the genome and/or transcriptome. This approach allows cognitive scientists to better understand the development of the "social brain," bypassing the need to compare distinct brain anatomies, which is often experimentally challenging, in favor of studying their more quantifiable molecular building blocks. In addition, brain-wide spatial localization of homologous genes or proteins supporting analogous behaviors across species can clarify how vastly different neuroanatomies can support analogous forms of social life [13,25,26]. Drawing from studies in social insects, variation in gene sequence and expression profile has suggested some degree of conservation between vertebrates and invertebrates in the neurogenomic architecture of social behavior, despite ~600 million years divergence from humans [24,27,28].

Sociability is thought to have a heritable component. In humans, for example, structural variation in the 7q11.23 genomic locus can dramatically affect the developmental trajectory of the social brain, causing either lifelong hypo- or hyper-sociability [7]. Such structural variants affect many aspects of an individual's neurobiology and physiology, making it challenging to pinpoint direct genetic effects on the propensity to engage in social interactions within a typical range of individual variation. Nevertheless, such findings indicate the existence of heritable molecular substrates that mediate the intensity of sociability.

One limitation in the study of vertebrate sociability is related to difficulties in tracking many individuals over long periods of time—a major challenge in relating genotype and social phenotype in humans [29]. It is similarly difficult to disentangle sociability from related traits, like locomotion, that contribute to overall activity rate: an individual may interact with others more because they are motivated to engage such interactions or because they are simply more physically active and, therefore, more likely to engage others by chance, or both. But, in some species, it is not possible to characterize individuals across multiple social phenotypic vectors. This issue has been largely overcome for social insects, as recent advances in automated tracking now allow for behavioral monitoring of each individual in a colony over long portions of their lives [30–34]. These advances permit finely resolved comparisons of social behavior, brain, and genome in a naturalistic setting.

Honey bees frequently engage in mouth-to-mouth sharing of liquid containing nutritive and signaling components in a stereotyped behavior called trophallaxis. Trophallaxis is thought to play important roles in colony nutrition and behavioral integration. Automated behavioral tracking of barcoded bees has shown consistent variation in trophallaxis frequency to be a robust proxy for several aspects of an individual's life history, from task specialization to disease status [5,35–37]. Moreover, the frequency at which a worker engages in trophallaxis interactions is linked to her occupation within the hive, suggesting a mechanistic coupling of the molecular factors that regulate behavioral state and those that shape an individual's predisposition toward social engagement [5].

More generally, trophallaxis acts as a "social glue" mechanism central to group cohesion in socially advanced insect societies [38], reflecting individual- and group-level tendencies toward reciprocal social interaction. In addition, variation in the duration of trophallaxis interaction in bees and face-to-face interactions in humans fall along similar heavy-tailed distributions, suggesting that universal principles may underlie the biology of social interaction networks [39]. Taken together, trophallaxis is a quantifiable analog to other measures of sociability across diverse taxa.

Although automated tracking of individuals engaged in trophallaxis has yielded new insights into the social structure of honey bee society, there are important gaps in the analysis of food-sharing sociability. These include genomics, transcriptomics, and movement dynamics associated with trophallaxis within the colony. Addressing these gaps promises to reveal deeply conserved neurobiological mechanisms that shape the social brain [32] as well as behavioral mechanisms associated with social bond formation, as in humans [40,41].

In honey bees, transcriptomic analyses have revealed provocative associations between variation in brain gene expression and variation in sociality. Shpigler and colleagues [24] found significant enrichment for autism-related genes in the brains of individual honey bees that were consistently unresponsive to task-related social stimuli. In a second study, these "socially unresponsive" bees also were found to have fewer and briefer trophallaxis interactions compared to age-matched individuals that were more responsive to task-related social stimuli [5]. These results established a link between variation in task performance and variation in sociability.

Here we report on behavioral, genomic, transcriptomic, and comparative genomic analyses of food-sharing sociability. First, we employed automated tracking of individually barcoded bees in colonies to reveal individual variation in this behavior. Second, we quantified movement kinematics to disentangle trophallaxis engagement frequency from overall activity rate and test hypotheses regarding how individual bees influence one another in terms of movement within the hive. Third, we followed these analyses with lab-based behavioral assays that robustly indicate task specialization. We then performed whole-genome resequencing and brain transcriptomic profiling on these extensively profiled individuals to identify genetic variants and brain gene expression profiles associated with variation in behavior (Fig 1a). We also localized the expression in the brain of one particularly interesting gene, *neuroligin-2* (*nlg2*), identified in both genetic and transcriptomic analyses. Finally, using comparative genomics, we explored similarities and differences in variation in food-sharing sociability, reward processing, and aggression in honey bees and across species as well.

## Methods

### Animals

All molecular and bioinformatic analyses were performed on bees collected as part of a previous study that addressed different questions [5]. The field- and lab-based behavioral experiments took place at the University of Illinois Bee Research Facility, Urbana, IL, USA, from August to September 2017. Briefly, we used adult worker bees collected from three colonies, each headed by a queen artificially inseminated with semen from a single male (drone, "SDI queen"). Individuals derived from these controlled matings have an average coefficient of relatedness of 0.75 due to haplodiploidy. This insemination strategy sharpens the resolution of genomic and molecular analyses by reducing intracolony variation without affecting the wild-type status of the colony. To obtain 0- to 24-h old (1-day-old) bees for age-matching, honeycomb frames containing late-stage pupae were removed from colonies, maintained in a dark incubator at 34°C and 50% relative humidity (RH), and ~500 individuals were gently swept off the frames the morning of each behavioral experiment. All colonies were filmed when the bees were 3–10 days old. On day 10 of the recording, bees were removed and prepared for laboratory-based behavioral assays, as described below.

We analyzed a total of 357 and 176 worker honey bees via DNA and RNA sequencing, respectively, with the latter being a subset of the former. Each analysis represents individuals from three colonies, "R2," "R8," and "R41," which respectively contributed n = 158, 126, and 73 bees for DNA analysis. For RNA sequencing, we leveraged lab-based behavioral assays that helped characterize several behavioral groups, including generalists (n = 25), foragers (n = 50), nurses (n = 28), guards (n = 32), and non-responders (n = 41).

### Automated behavioral tracking, image processing, and trophallaxis detection

Single-sided, glass-walled observation hives, each containing a colony with barcoded bees, were set up as previously described [5,31,35,36]. Individual bees were cold-anesthetized and a barcode was applied to the dorsal thorax using a small amount of cyanoacrylate glue. Barcoded bees were then gently transferred to an observation hive containing one honeycomb frame provisioned with honey (top 18 rows, ~200 mg per cell, ~330 g in total) and pollen paste (next six rows, ~100 mg per cell, ~45 g total; pollen paste was made from 45% honey, 45% pollen and 10% water). A naturally mated queen, unrelated to the workers, was lightly anesthetized with carbon dioxide, barcoded, and allowed to recover surrounded by workers in the observation hive.

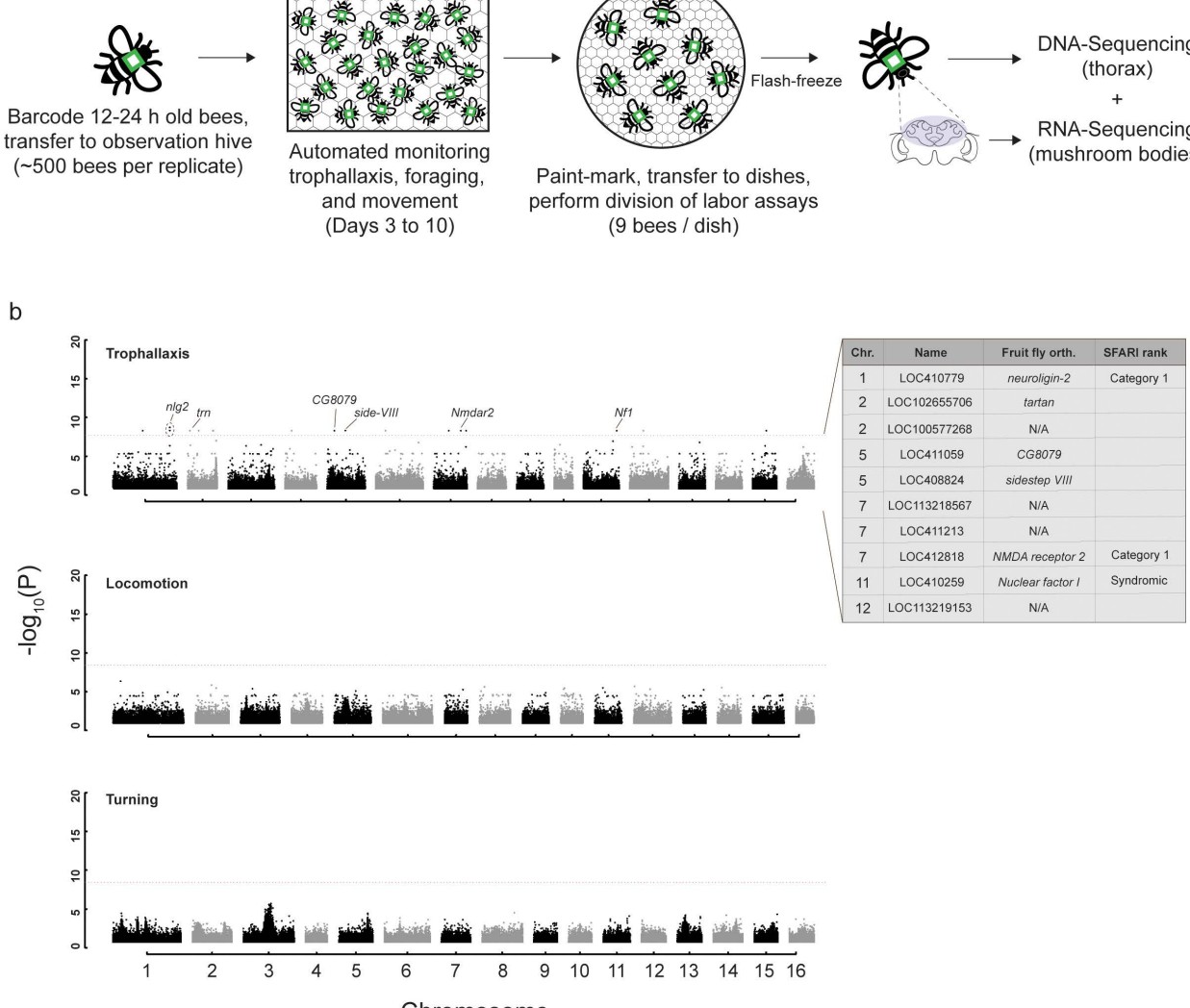

**Fig 1. Experimental design and genomic correlates of honey bee trophallaxis frequency. (a)** For each replicate trial, a mix of ~500 bees from two source colonies was barcoded and transferred to a single honeycomb frame for automated monitoring of trophallaxis, foraging, and movement behaviors. After 10 days, bees were transferred to a petri dish and subjected to aggression and affiliative care assays. Immediately following behavioral assays, bees were flash-frozen and stored until the thorax and brain were dissected for DNA and RNA sequencing, respectively. Our final sample sizes for whole-genome resequencing and mushroom body RNA sequencing were 357 and 176, respectively, and a total of three source colonies were used across two replicate trials. **(b)** (Top) Genome-wide association study (GWAS) identified 18 single nucleotide polymorphisms (SNPs) significantly associated with variation in the frequency of trophallaxis (food-sharing) interactions, 12 of which were localized to the introns of gene models in the honey bee genome (HAv3.1). In vertebrates, variation in the expression or structure of three of these genes, *neuroligin-2*, *NMDA receptor 2*, and *Nuclear factor I* is associated with autism per the Simons Foundation Autism Research Initiative (SFARI) database. (Middle and bottom) In contrast, we did not identify any SNPs associated with variation in locomotion or turning kinematics, movements that contribute to, but do not fully encompass, trophallaxis interactions. Dashed red line indicates significance threshold (Bonferroni-corrected *P*-value < 0.05). Code and data underlying Fig 1b can be found in S1 Table and https://doi.org/10.6084/m9.figshare.29845490.

Observation hives were then transferred to a dark room connected to the outside by a foraging port that was kept closed for the first 2 days of the experiment to prevent young bees with undeveloped flight muscles from exiting the colony. For recording, we used LED lights (Smart Vision Lights, Muskegon, MI, USA) to illuminate the honeycomb with

infrared light, which the bees cannot see. Whole-colony images were captured at a rate of 1 image/s using a Prosilic GT6600 machine vision camera (Allied Vision, Exton, PA, USA). To ensure sharp images, the observation hive glass window was gently replaced daily with a clean one without disturbance to the colony.

Hive images were processed as previously described [5,31,42]. Briefly, images were resized and sharpened to improve barcode detection rate, and each bee's location and orientation were inferred using custom software [31], which also filtered any barcode that was unreadable or resulting from a spurious detection. We then identified trophallaxis events by recording when two bees were close (1.7–7.4 mm apart), faced each other, and a convolutional neural network (CNN) confirmed that one bee had inserted her extended proboscis ("tongue") between the open mouthparts of the other bee [5,42]. These "raw" trophallaxis events were merged if they were consecutive and < 1 min apart. Merged interactions < 3 s were discarded, as such short interactions may involve no liquid transfer [43], and interactions > 3 min were removed as spurious detections, since bees do not usually interact this long [44,45]. While a bee cannot be identified if her tag is missing (an event too rare to meaningfully quantify), the design of the observation hive makes it challenging for bees to walk over one another due to the narrow margin of space between the glass and honeycomb. A tag also cannot be detected if a bee is checking a cell or feeding a larva, but trophallaxis between adults does not occur in these instances.

The detection of trophallaxis events using this automated tracking system has been subjected to rigorous ground-truthing via manual annotation, demonstrating that our approach identifies 89% of all trophallaxis interactions with a true positive rate of 90% [31,42]. Moreover, a comparison of field-based trophallaxis observations to those automatically detected by this system shows significant concordance between the behavior observed as we have done here and the behavior observed under more natural conditions [35].

## Bee kinematics quantification

We quantified bee movement as the instantaneous linear (locomotion) and angular (turning) speeds of individuals. Linear speed was defined as the Euclidean distance between a bee's barcode center at time points t and t + 1, divided by the image rate r. Angular speed a was defined as the unsigned angle between a bee's barcode orientation vector at t and t + 1, divided by r. Mean speeds were calculated by averaging these quantities across time. If a bee was not detected at time point t, we recorded no speed measurements for the image captured at time point t + 1. We also recorded no speed measurement if a bee moved less than the width of a honeycomb cell (4.9 mm) and turned less than from one cell corner to the next (60°). These thresholds were applied to differentiate actual locomotion from the small movements that a bee may perform while she is stationary (e.g., during allogrooming and other behaviors).

## Social influence analysis

To measure the extent to which the movement of one bee influenced the movement of another bee, we computed the information partial directed coherence (iPDC) [46], employing the AsympPDC package, https://www.lcs.poli.usp.br/~baccala/pdc/CRCBrainConnectivity/ [47]. iPDC measures the directed coherence between two or more time-series, enabling estimation of the bidirectional transfer of movement influence-related information. As previously done [5], we focused on the last 2 days of the recording and limited analysis to periods characterized by proximal interactions (i.e., when two bees were physically close to each other for a minimum amount of time), since it is unlikely that two bees located far apart in the hive would influence each other's movement directly. We chose to focus our analyses and others, as described below, on this time window, as individual-level trophallaxis interaction frequency is highly stable over time, especially after the first few days of life [5].

To identify periods of proximal interaction, we computed for each image the Euclidean distance $R$ between a focal bee and all other bees, using the $X$ and $Y$ coordinates of their barcodes. Given a focal bee, we defined *neighbor* as another bee whose distance $R$ to the focal bee was equal or lower than a radius $R_{max}$ for a duration $L$ equal or higher than a minimum duration $L_{min}$ ($R \leq R_{max}$, $L \geq L_{min}$). We chose $R_{max} = 2$ cm for being approximately twice the length of a bee's body

and because previous studies used this value as a measure of proximal interaction in honey bees (Wild and colleagues, 2021). We chose $L_{min} = 30$ images as a minimum duration of the proximity interaction in order to have sufficient time-series samples for computing iPDC. In addition, since our aim was to measure social influence during movement, we excluded images where either one of the two bees' speed was very low as well as images where the two bees were engaging in trophallaxis, during which animals are relatively still. If a focal bee interacted with the same neighboring bee multiple times satisfying the above criteria, we kept data from the first interaction only. In this manner, the identity of the neighbor varied with each proximity interaction. The number of proximity interactions varied across focal bees, depending on the number of unique neighbors they encountered, which could be from the same or different source colony.

We retrieved bees' $X$ and $Y$ coordinates during each proximity interaction and built an input dataset to which iPDC was applied. In the dataset, each interaction is represented as a bivariate time-series of at least 30 time points (30 s), where the first time-series is the focal bee's $X$ or $Y$ coordinate and the second time-series is its neighbor's $X$ or $Y$ coordinate, respectively. Next, we fit a vector autoregressive model (VAR) to the time-series in each interaction and computed from it the iPDC spectra, from the focal bee to the neighbor and vice versa. iPDC spectra were then averaged across interactions to obtain a single iPDC spectrum for each focal bee, in both directions. Finally, the averaged iPDC spectra were integrated across all frequencies into information flow ($I_{flow}$), as adapted from equation 8 in (Takahashi and colleagues, 2010):

$$ I_{flow} = -\frac{1}{fs} \cdot \int_{0}^{fs/2} log2\left(1 - iPDC(f)\right) \ df $$

where $fs$ is the sampling rate.

We, thus, obtained a scalar value representing the magnitude of the influence between the focal bee and its neighbors in units of information transfer (bits).

## Behavioral state identification via laboratory-based assays and foraging detection

Behavioral states related to natural colony division of labor (guards, nurses, foragers, generalists, and non-responders) were studied under controlled laboratory conditions as previously described [5]. Briefly, at the conclusion of the recording portion of the experiment, the glass observation window covering the hive was replaced with a Plexiglas window containing resealable portholes, and groups of nine bees were gently removed, paint-marked with a unique color applied to the dorsal thorax (Testors Paint, Rockford, IL, USA), and transferred to a vertically oriented 100 x 200 mm Petri dish (Thermo Fisher Scientific, Waltham, MA, USA). Dishes contained a tube of honey (~1.2 ml), 50% sucrose solution (2 ml), and a pollen ball (70% pollen, 30% sucrose solution described above). Bees were given at least 60 min to acclimate to normal fluorescent lighting prior to the start of the behavioral assays.

Established laboratory assays were used to identify aggression, which reflects guarding behavior [48,49] and affiliative caregiving, which reflects nursing behavior [50,51]. Aggression was measured by subjecting groups of bees to a 5-min interaction with a foreign bee ("intruder") and quantitating aggressive biting and stinging interactions. Affiliative caregiving was measured by exposing groups of bees to a larva for 5 min and quantitating observations of caregiving interactions like food provisioning and inspection. Each group was exposed to both stimuli, and "guards" and "nurses" were identified based on consistent (≥ 20–30 s) observations of biting and stinging or larval feeding, respectively. We also noted instances of fanning, wax-building, and vibration-signaling behaviors (following descriptions in [21]) to give a more comprehensive depiction of each individual's behavioral state.

Affiliative care assays were performed in a controlled environment that mimicked the interior of the hive (34°C and 50% RH) under ambient lighting. Aggression assays were performed in a cooler room (28°C) to approximate outdoor temperatures at the hive entrance, where defensive aggression by guards against intruders is most likely to occur. Each dish of nine bees was tested in both assays in a random order with the second assay performed 60 min after the first.

Immediately following the second assay, all bees were flash-frozen in liquid nitrogen, their barcodes removed for identification purposes, and stored at −80°C prior to DNA and RNA analysis.

Because the barcoded colonies were initially composed only of 1-day-old bees, "precocious" foraging occurred during the observation period from days 3 to 10 [52]. Foragers were identified via automated monitoring of flight in and out of the hive with an entrance monitor attached to the outer portion of the entrance tube connected to the observation hive [35]. Although foraging typically occurs toward the end of a honey bee's lifetime, in colonies such as these that lack older bees, a subset will exhibit "precocious" foraging to serve the colony's nutritional needs [22,53]. The entrance monitor contained a small enclosure with a simple maze for slowing bees down to facilitate image capture. A Raspberry Pi Camera Module v1.3 was mounted above the enclosure and separated from the maze by a removable glass window. The camera recorded.mjpg videos at a temporal resolution of three images/s from the hours of 07:00–19:00 h CST, automatically adjusting for changes in light conditions throughout the day.

Videos of bee activity were first converted to still images using ffmpeg (https://ffmpeg.org/) and then processed to detect barcoded bees, as previously described [36]. Raw incoming and outgoing "passes" were merged by vectoring the bee's displacement toward or away from the hive entrance. To be labeled a forager, we required bees to be detected taking at least six trips, with more than three trips per day, for any 2 days of the experiment, and at least 25% of these trips had to be made during peak foraging hours (10:00–15:00 h CST). This is consistent with honey bee foraging behavior under natural conditions [54].

Bees that did not respond to behavioral stimuli during the lab assays and were not observed to forage when in their observation hive were labeled "non-responders" [24]. Bees that showed mild responsiveness (i.e., < 20 s interaction) and displayed some other behaviors (fanning, wax-building, and vibration-signaling behaviors) were referred to as "baseline" bees for which the behavioral state could not be robustly identified but were clearly responsive to stimuli. Bees that exhibited two or more behaviors (i.e., brood care, aggression, and/or foraging) were "generalists."

## DNA and RNA sequencing and analysis

All DNA and RNA sequencing data were newly generated for this study from the individuals analyzed behaviorally as described above. Raw and processed genomic and transcriptomic data are publicly available on NCBI under Bioproject PRJNA1237396.

### DNA sequencing

DNA was extracted from the thorax of 391 individual bees using the Gentra Puregene Tissue Kit (QIAGEN, Germantown, MD, USA) as previously described [55]. We used the thorax for DNA analysis because it reliably yields large amounts of DNA and, because this study does not explore somatic mutation in the brain, we expected the genome of the thorax to be generally representative of individual genotype, further allowing us to save the brain for behavioral transcriptomic analyses, as described below. Shotgun genomic libraries were prepared using the Illumina DNA (Nextera Flex) sample preparation kit, pooled, quantitated via qPCR, and sequenced on one S4 lane for 151 cycles from both ends of the fragments on an Illumina NovaSeq 6000. Raw fastq files were generated and demultiplexed using the bcl2fastq v2.2 conversion software (Illumina, San Diego, CA, USA).

### Mushroom body RNA sequencing

We sequenced RNA from 176 individuals, a subset of the larger set used for DNA sequencing. These individuals were selected because they each had a definable behavioral state other than "baseline," following our above-described assignment strategy. We did not sequence "baseline" bees, as these individuals did not forage or robustly engage social stimuli in the dish assays but showed higher levels of engagement than non-responders, in addition to other well-known natural

behaviors like wax-building or fanning. As we could not reliably characterize these individuals in behavioral terms, molecular profiling would not be as meaningful as for other groups.

The mushroom body (MB) contains several hundred thousand neurons, glia, and supportive brain cells, such that sufficient quantities of RNA can be extracted for transcriptomic analyses at the level of the individual bee. We performed MB dissection and RNA extraction on individual brains without pooling across bees, following previously established methods optimized for honey bee neurobiology [5,56]. We chipped a small window on the anterior surface (frons) of flash-frozen bee heads immersed in a dry ice/ ethanol bath before submerging the entire head in RNAlater-ICE Frozen Tissue Transition Solution (Thermo Fisher Scientific) overnight at −20°C. The brain was then fully dissected on wet ice and the MB was removed and stored at −80°C until RNA extraction.

We used the PicoPure RNA Isolation Kit (Thermo Fisher Scientific) with DNase treatment (Invitrogen, Carlsbad, CA, USA) as has been previously described [5,24] and quantified via QuBit fluorometer (Thermo Fisher Scientific). RNA sequencing libraries were prepared with the KAPA Hyper Stranded mRNASeq Sample Prep Kit (Illumina); libraries were pooled, quantitated via qPCR, and sequenced on an S4 lane for 151 cycles from both ends of the fragment on an Illumina NovaSeq 6000.

## Bioinformatic analyses

**DNA alignment, variant calling, and Genome-Wide Association Study (GWAS).** Whole-genome resequencing yielded at least 40,000,000 paired-end reads per sample (mean [$\mu$] ± standard deviation: $\mu = 64,900,620 \pm 9,035,972$). After adaptor trimming, raw reads were aligned to the honey bee genome, Amel_HAv3.1 [57], using the Burrows–Wheeler algorithm, and resulting alignment files were sorted and deduplicated. The Samtools "flagstat" report indicated > 99% mapping rates for samples to the honey bee genome, and deduplication of sequenced reads suggested a duplication rate of < 20% ($\mu = 16.8 \pm 0.012\%$). Thus, each sample was sequenced at an average depth of 30x, indicating a high-quality dataset suitable for GWAS.

Variant calling was performed using the Sentieon DNAseq workflow (https://support.sentieon.com/manual/DNAseq_usage/dnaseq/). Following indel realignment and variant calling, which resulted in a VCF file for each of our samples, genotyping was conducted across all samples to generate a single, multi-sample VCF file. We then removed indels, multiallelic variants, and variants localized to unplaced scaffolds or mitochondrial genes. The remaining biallelic SNPs were filtered using joint quality measures and overall representation in the dataset, utilizing missing and low-coverage calls in the multi-sample VCF file. No sample showed excessive missing reads. Sequenced genomes were then phased and imputed with Beagle (v5.1) [58,59]. Linkage disequilibrium (LD) pruning was performed using the snpgdsLDpruning command in the R package SNPRelate [60] using the D' metric with an LD threshold set to 0.3 and mean allele frequency set to 0.2. The resulting ~2.5 million SNPs were used for downstream analyses.

We derived a genetic relatedness matrix (GRM) using the "pcrelate" function in GENESIS [61] to account for variation based on kinship, as has been previously done in honey bees [55]. Bees that did not cluster with their colony of origin, and/or bees that were later found to have unreadable barcodes (and, therefore, unreliable identifying information), were excluded from analysis, leaving us with a final sample size of 357 individuals for DNA analysis.

For individual-level phenotypic association, we utilized the quasi-likelihood approximation in GENESIS, which scores and assigns a *P*-value to each SNP, thus predicting the relevance of the association between a given SNP and phenotype for each genotype. We used GENESIS to identify SNPs significantly associated with trophallaxis frequency, kinematic metrics (locomotion and distance traveled), and social influence exchange, all averaged over the final 2 days of the recording experiment, as described in the above text. Following a previously identified, significant link between division of labor and trophallaxis frequency [5], we excluded behavioral state as a predictor variable to avoid collinearity in our GWAS model. For resulting SNP sets, polygenic risk scores (PRS) were calculated by summing effect size-weighted SNPs associated with a given phenotype via GWAS.

**Brain MB RNA sequencing alignments and detection of differentially expressed genes.** Each sample yielded at least 13,000,000 paired-end reads ($\mu = 17,836,571 \pm 2,286,238$). After demultiplexing, raw reads for each sample were aligned to the honey bee genome, Amel_HAv3.1 [57] using STAR v2.7.3a [62] and reads were counted using the *featureCounts* function in the Subread v2.0 package [63]. We identified the presence of Deformed wing virus (DWV), a virus that is naturally widespread in honey bee apiaries and commonly found in the brain [49,64] by also aligning raw reads to the DWV genome. All linear models for identifying differentially expressed genes (DEGs) included percentage of total reads aligned to the DWV genome as a covariate, as well as colony of origin, in order to control for unwanted variation due to natural viral presence or latent genetic variation across colonies. We observed a unique mapping rate of $84.5 \pm 0.0845\%$ against the honey bee genome, and an average of $65.7 \pm 0.066\%$ of these reads were uniquely mapped to a gene model.

Raw gene counts for 12,332 genes were analyzed with DESeq2; no genes were filtered based on expression values, which has a negligible effect on differential expression testing [65]. Briefly, gene counts were normalized to library size to account for variation in sequencing depth across samples, and significant logarithmic fold-changes associated with predictor variables were determined based on the Wald test. For principal components analysis, a variance-stabilizing transformation was first applied to library size-normalized count matrixes.

We first identified DEGs associated with sociability by regressing MB gene expression using trophallaxis frequency, averaged over the last 2 days of the experiment [5], as a continuous predictor. Next, we calculated DEGs associated with each behavioral state (generalist, forager, guard, nurse, or non-responder) by comparing each state to expression levels averaged across the remaining four groups. Each DEG list was generated via the Wald test and corrected for multiple testing using the Benjamini–Hochberg method, FDR < 0.05. Gene Ontology (GO) enrichment analysis was performed by converting honey bee gene identifiers to one-to-one orthologs in *Drosophila melanogaster* via a previously published reciprocal best-hit BLAST [64], detecting term enrichment in GOrilla [66], and visualizing results in GO-Figure [67]. Gene list overlap analyses were performed using the GeneOverlap package (https://bioconductor.org/packages/GeneOverlap) with Bonferroni corrections for multiple testing.

***Spatial analysis of** neuroligin-2 **gene expression.*** *In situ* hybridization was performed in January 2023 to provide additional information on one of the genes identified in DNA and RNA analyses that emerged as a particularly strong sociability-associated candidate gene, *neuroligin 2* (*nlg2*). One-day-old ("callow") individuals were collected from a honeycomb frame containing brood from a colony kept indoors at the Bee Research Facility. To control for social environment, bees were kept in a small Plexiglas cage at 35°C and 50% RH until whole-brain dissections were performed at 7 or 14 days of age. Sampling at days 1 (callow), 7, and 14 allowed us to capture the trajectory of behavioral maturation, as callow workers are entirely colony-bound, 7-day-olds specialize in in-hive tasks like nursing and comb maintenance, and 14-day-olds are prepared to engage in out-of-hive activities like nest defense and foraging.

We followed an established protocol (https://www.protocols.io/view/hcr-rna-fish-protocol-for-the-whole-mount-brains-o-bzh5p386) with minor modifications. Brains were dissected in phosphate-buffered saline (PBS) and fixed in 4% paraformaldehyde (in 1X PBS with Triton-X100; PBS-T) overnight at 4°C. Fixed brains were washed three times in PBS with Tween (Thermo Fisher Scientific), dehydrated in serial methanol/PBST washes (25%, 50%, 75% and 100%), and stored at −20°C until use.

Brains were rehydrated in the reverse methanol/PBS-T serial washes (75%, 50%, 25%) and washed in PBS-T. Brains were pre-hybridized in hybridization buffer (Molecular Instruments, Los Angeles, CA, USA) without probes for 1 h at 37°C. Next, we combined 4 µL of 1 µM hybridization chain reaction (HCR) detection probes targeting the honey bee *nlg2* gene (Molecular Instruments) with 250 µL hybridization buffer, and incubated each brain in this solution overnight at 37°C. The following day, brains were washed in preheated probe wash buffer (Molecular Instruments) 5 times, 10 min per wash, at 37°C, and then in 5X saline-sodium citrate twice for 5 min per wash. Washed brains were pre-amplified in amplification buffer (Molecular Instruments). Signal amplification hairpin oligos (B1-H1-488 and B1-H2488, from Molecular Instruments)

were first denatured at 95°C for 90 s and annealed by cooling on wet ice for 30 min. Amplification hairpins were used at a concentration of 6 μL per 100 μL amplification buffer. Brains were incubated in the amplification probe solutions overnight, followed by two washes in probe wash buffer containing DAPI (Thermo Fisher Scientific, R37606). The samples were washed in PBS and cleared with RapiClear 1.49 (SunJin Lab Co, Hsinchu City, Taiwan).

Confocal imaging was performed with a Zeiss LSM 900 inverted confocal microscope (Carl Zeiss, Oberkochen, Germany) with 405 and 488 nm excitation wavelengths. The 20X/0.8 and 25X/0.8 objectives were used for scanning the whole brain in the tile-scan mode. The Plan-Apochromat 25X/0.8 and 63X/1.4 oil objectives were used for scanning MB sub-regions. Images were viewed and processed with the ZEN microscopy software (Zeiss).

## Results

### Genome-wide association study (GWAS) on trophallaxis frequency and movement kinematics

We first asked if there are genomic correlates of trophallaxis frequency, utilizing automated tracking of trophallaxis interactions performed across two replicate trials with experimental colonies composed of bees from three source colonies: R2, R8, and R41. Colony 1 was composed of bees from R2 and R41 and Colony 2 was composed of bees from R2 and R8. We generated a ~2.5 million SNP-set from whole-genome sequencing of 357 bees that we then correlated with trophallaxis frequency. Trophallaxis frequency was quantified as the average number of trophallaxis interactions per bee across the last 2 days of recording, as in [5].

This analysis identified 18 single nucleotide polymorphisms (SNPs) localized across nine chromosomes (Bonferroni-corrected $P < 0.05$; Fig 1b). Eleven of these SNPs localized to introns of honey bee gene models [68], one localized to a predicted noncoding RNA (LOC113219153), and the remaining six were found outside of genes (Fig 1b and S1 Table). Among the 11 intronic SNPs, two were found in the gene *neuroligin-2*, two in *glutamate receptor ionotropic, NMDA 2B,* which had robust one-to-one orthology (detected via a previous reported reciprocal best hit BLAST [64]) with the *D. melanogaster* genes *neuroligin-2* and *NMDA receptor 2*, respectively. A single SNP was found in the following genes, with one-to-one orthologs in *D. melanogaster* given in parentheses: *Angiogenic factor with G patch and FHA domains 1* (CG8079), *CCR4-NOT transcription complex subunit 6-like* (no hit in *D. melanogaster*), *nephrin* (*sidestep VIII*), *discoidin domain-containing receptor 2* (no hit in *D. melanogaster), nuclear factor 1 X-type* (*nuclear factor 1*) and uncharacterized genes LOC102655706 (*tartan*) and LOC100577268 (no hit in *D. melanogaster*). We also performed a "conditional" reciprocal best hit blast [69], which accounts for evolutionary divergence between species, between coding sequences of honey bee and human genes, but did not find a hit for *nlg2*. However, a BLASTP analysis yielded many significant (e-value ≤ $10^{-60}$) hits with low (≤ 30%) identity between honey bee *nlg2* and several members of the human *neuroligin* gene family, suggesting deep conservation of this family across distinct animal lineages.

In a comparative bioinformatic analysis, n*euroligin-2* (*nlg2*) and *NMDA receptor 2* (*nmdar2*) were found to be Category 1 ("high confidence") genes in the Simons Foundation Autism Research Initiative (SFARI) gene database (https://gene.sfari.org/), meaning they have been clearly implicated in human Autism Spectrum Disorder. Similarly, *nuclear factor 1 X-type* (*Nf1*) is characterized as a "Syndromic" autism gene, meaning its dysfunction is associated with the development of autistic traits that are not typically associated with a conventional autism diagnosis.

We performed the same GWAS analysis for two other behaviors: locomotion and turning. Like trophallaxis, these behaviors require fine motor control and, therefore, contribute to trophallaxis interactions, but without obvious social components. (Fig 1b). In contrast to our findings for trophallaxis frequency, GWAS for these two behaviors did not result in the detection of significantly associated SNPs.

### Neuroligin-2 (nlg2) and colony sociability

Of the three source colonies assayed in this study, individuals from source colony R41 were found to all be homozygous for the alternative allele (i.e., containing alleles that were different from the reference locus in the honey bee genome) for

the majority of identified SNPs (Figs 2a and S1). In contrast, individuals from source colonies R2 and R8 were predominantly heterozygous for the more prevalent reference allele. This result prompted us to investigate the genetics of colony-level variation in behavior.

Trophallaxis frequency significantly varied with source colony (ANOVA: $F_{(2,354)}$ = 3.165, $P$=4.24e-16), and bees from source colony R41 displayed significantly lower trophallaxis rates than those from source colony R8 (Tukey HSD post-hoc test, $P$=0.034), but not R2 ($P$=0.23; Fig 2b). In addition, R2 individuals were significantly less social when co-housed with R41 individuals in Colony 1 than with R8 individuals in Colony 2 (Wilcoxon rank-sum test: W=6833.5, $P$=4.17e-11), further implicating R41 as significantly less sociable than the other genotypes.

While individuals from source colony R41 demonstrated the lowest levels of trophallaxis frequency, they were not the slowest moving relative to bees from the other two source colonies. Locomotion was significantly associated with source colony (ANOVA: $F_{(2,349)}$ = 12.52, $P$<4.24e-16), and each colony significantly differed from the others in this measurement (Tukey HSD post-hoc test, $P$<0.005 for all comparisons). However, movement dynamics could not explain the observed variation in trophallaxis rate (Fig 2c).

## Social influences on movement in the hive

To explore bee social interactions further, we measured the extent to which a given bee's movement influenced, and was influenced by, the movement of a nestmate. We implemented iPDC, which allows for the measuring of bidirectional influences between signals [46]. To this end, we identified periods of proximal interaction (i.e., when two bees were physically close to each other), retrieved the $X$ and $Y$ coordinates of each bee relative to the upper-left corner of the hive, and quantified iPDC from the coordinate time-series of one bee to the coordinate time-series of its neighbor and vice versa. From iPDC, we then computed information flow ($I_{flow}$) (Fig 2d-f; see also Methods). Higher $I_{flow}$ values indicate higher information transfer from the past location of a bee to the current location of its neighbor, and we used this metric as measure of social influence. $I_{flow}$ was approximately normally distributed across bees (Fig 2g; $I_{flow\ bee->neighbor}$ μ=0.0261 bits, σ=0.0033; $I_{flow\ neighbor->bee}$ μ=0.0261 bits, σ=0.0029) and the values observed are consistent with those reported for other animal species when assessing social influence in movement dynamics [70]. We then asked whether bees of different colonies exhibited different levels of social influence. Information flow significantly varied with source colony (Figs 2h and S2; one-way ANOVA $I_{flow\ bee->neighbor}$: $F_{(2,352)}$ = 12.26, $P$=7.11e-06; $I_{flow\ neighbor->bee}$: $F_{(2,352)}$ = 7.1, $P$=9.51e-4). Post-hoc comparisons revealed no significant difference between colonies R2 and R8 (Tukey HSD $I_{flow\ bee->neighbor}$: $P$=0.58; $I_{flow\ neighbor->bee}$: $P$=0.17). In contrast, consistent with the trophallaxis results, bidirectional $I_{flow}$ was lowest for individuals from colony R41 compared to individuals from colony R2 (Tukey HSD $I_{flow\ bee->neighbor}$: $P$=1.37e-4; $I_{flow\ neighbor->bee}$: $P$=0.045) and R8 (Tukey HSD $I_{flow\ bee->neighbor}$: $P$=7.6e-6; $I_{flow\ neighbor->bee}$: $P$=5.69e-4), suggesting that R41 bees show reduced influence on, and from, their neighbor's movement during time of social proximity.

As for trophallaxis frequency and kinematic metrics, we similarly performed a GWAS using social influence as the focal trait. Neither $I_{flow\ bee->neighbor}$ nor $I_{flow\ neighbor->bee}$ were associated with genotypic variation, and no SNPs were detected at Bonferroni-corrected $P$<0.05. However, regressing the PRS calculated for the trophallaxis frequency SNPs on social influence metrics identified significant, negative relationships for both $I_{flow\ bee->neighbor}$ and $I_{flow\ neighbor->bee}$ (Fig 2i, Spearman's rank-order correlation, −0.29 and −0.18, respectively; $P$<0.001 for each test).

## *Mushroom body (MB) transcriptomic profiles and trophallaxis* frequency

We next asked how trophallaxis frequency is related to gene expression, focusing on the MBs, a higher-order center of sensory integration, decision-making [14], and social regulation [50]. Unlike GWAS, transcriptome profiling generates arrays of genes that associate with a particular continuous or discrete phenotypic variable. Deriving lists of genes with expression values correlated with trophallaxis rate, for example, allows us to test for statistically significant overlap between our findings and previous studies of other behaviors, such as reward processing. We used RNA

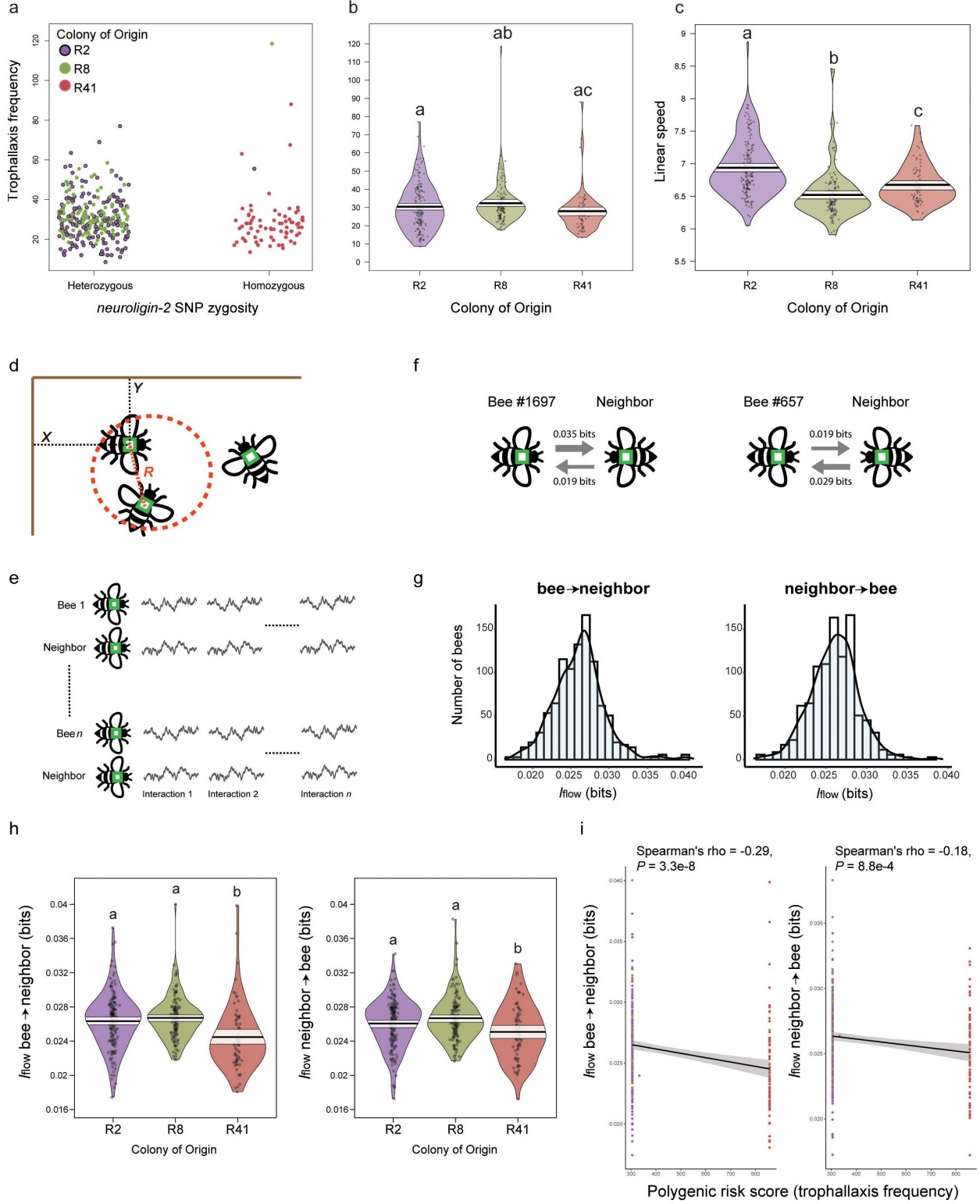

**Fig 2. Genotype-specific variation in motor synchrony. (a)** A single source colony, R41, was found to be homozygous for *nlg2* as well as the majority of trophallaxis-associated SNPs, while other colonies in this experiment contained the more prevalent reference allele. **(b)** Colony R41 also

demonstrated the lowest levels of trophallaxis frequency, yet (c) was not the slowest-moving colony. **(d)** Images of proximal interaction were selected where two bees were physically close to each other ($R \leq 2$ cm, where $R$ is the Euclidean distance between the two bees' barcodes). The $X$ and $Y$ coordinates of the barcode represent a given bee's position in Euclidean space. **(e)** Drawing representing the dataset to which information partial directed coherence (iPDC) was applied. Each proximal interaction (interaction 1, … interaction $n$) is represented as a bivariate time-series of at least 30 time points (30 s), where the first time-series is the focal bee's $X$ or $Y$ coordinate and the second time-series is its neighbor's $X$ or $Y$ coordinate, respectively. For each focal bee, the identity of the neighbor varied across interactions. iPDC was calculated within each paired time-series and averaged across interactions to subsequently compute information flow in both directions. **(f)** Example of two bees with different levels of social influence. Arrow thickness is proportional to information transfer. On average, Bee #1697 influenced the movement of its neighbor with $I_{flow\ bee \to neighbor} = 0.035$ bits, and was influenced by the movement of its neighbor with $I_{flow\ neighbor \to bee} = 0.019$ bits. **(g)** Distributions of information flow across bees, representing the magnitude of movement influence-related information. **(h)** As for trophallaxis frequency, R41 bees were also least likely to influence, or be influenced by, other bees in their experimental cohort in terms of motor synchrony. Violin plots are constructed as follows: points represent raw data, solid black lines represent the mean, pale white shading above and below the mean represent a 95% confidence interval, and plot shape represents a smoothed density curve outlining the distribution of raw data. Letters above violin plots represent significance ($P$-value < 0.05) from a between-group Tukey post-hoc analysis following a one-way ANOVA. **(i)** Social influence was also negatively correlated with a per-bee polygenic risk score (Spearman's rank correlation, $P$-value < 0.0001 for each correlation test); polygenic risk score was calculated by summing effect size-weighted SNPs associated with trophallaxis frequency via GWAS. Code and data underlying Fig 2 can be found in S1 Table and https://doi.org/10.6084/m9.figshare.29845490.

sequencing (RNA-Seq) to profile the MB transcriptome of a subset of bees included in the GWAS, selecting individuals that were reliably identified as generalists, foragers, guards, nurses, or non-responders, following criteria from previous studies [5,24,48].

Non-responders were transcriptionally distinct from generalists, foragers, guards, and nurses, as in [24]; (S2-S6 Tables). A GO enrichment performed on genes that were upregulated in all responders (i.e., any behavioral group other than non-responder) compared to non-responders identified "chemosensory behavior" and "learning or memory" as enriched in responders whereas, for genes upregulated in non-responders, only a small number of less specific metabolic terms were identified as enriched (Fig 3a). Along with foragers, non-responders had a larger number of upregulated genes than other behavioral groups. In addition, principal component analysis also showed a strong effect of colony of origin (Supp. Fig 3a). Foragers are extremely physically active and must learn to navigate complex landscapes outside of the hive whereas non-responders are characterized by a lack of observable attendance to social stimuli or engagement of foraging activity. These results suggest that the non-responder phenotype in honey bees also has complex molecular underpinnings [5,24].

In light of these results, we sought to further characterize the non-responder phenotype by comparing our results to a previous brain transcriptomic analysis of foragers and soldiers [71], a class of forager-aged bees that show higher aggression levels relative to foragers. Soldiers are the first bees that attempt to sting invertebrate and vertebrate threats to the colony rather than contribute to foraging efforts [71]. We identified significant overlap between both generalists ($P = 3e-03$; all $P$-values for overlap tests are Bonferroni-corrected) and foragers ($P = 1e-52$) from this study with foragers from Traniello and colleagues (2023), which is unsurprising given that many generalists, by our definition, perform foraging alongside other tasks. The stronger enrichment between forager brain gene expression profiles across both studies is likely reflective of a more direct comparison as foragers, in this study, are specifically defined as not performing any activities besides foraging (S3b Fig). Nurses and guards, expectedly, did not overlap with forager or soldier MB gene expression profiles from Traniello and colleagues (2023), but we did identify a significant overlap of genes that characterize non-responders and soldiers ($P = 3e-06$).

Finally, we identified 920 and 785 genes with expression values that positively or negatively correlated with individual differences in sociability, respectively (S7 Table). Several signaling-related genes like *neurexin-4*, *neural-cadherin*, *synaptogyrin*, *synaptojanin-1*, and *synaptotagmins 4, 7,* and *10* were positively associated with trophallaxis frequency. GO enrichment analysis identified distinct terms for each set of genes, with positively associated genes enriched for several regulatory processes in the context of cellular communication and signaling and negatively associated genes enriched for processes associated with translation and macromolecule biosynthesis (Fig 3b). In contrast to our GWAS results, we

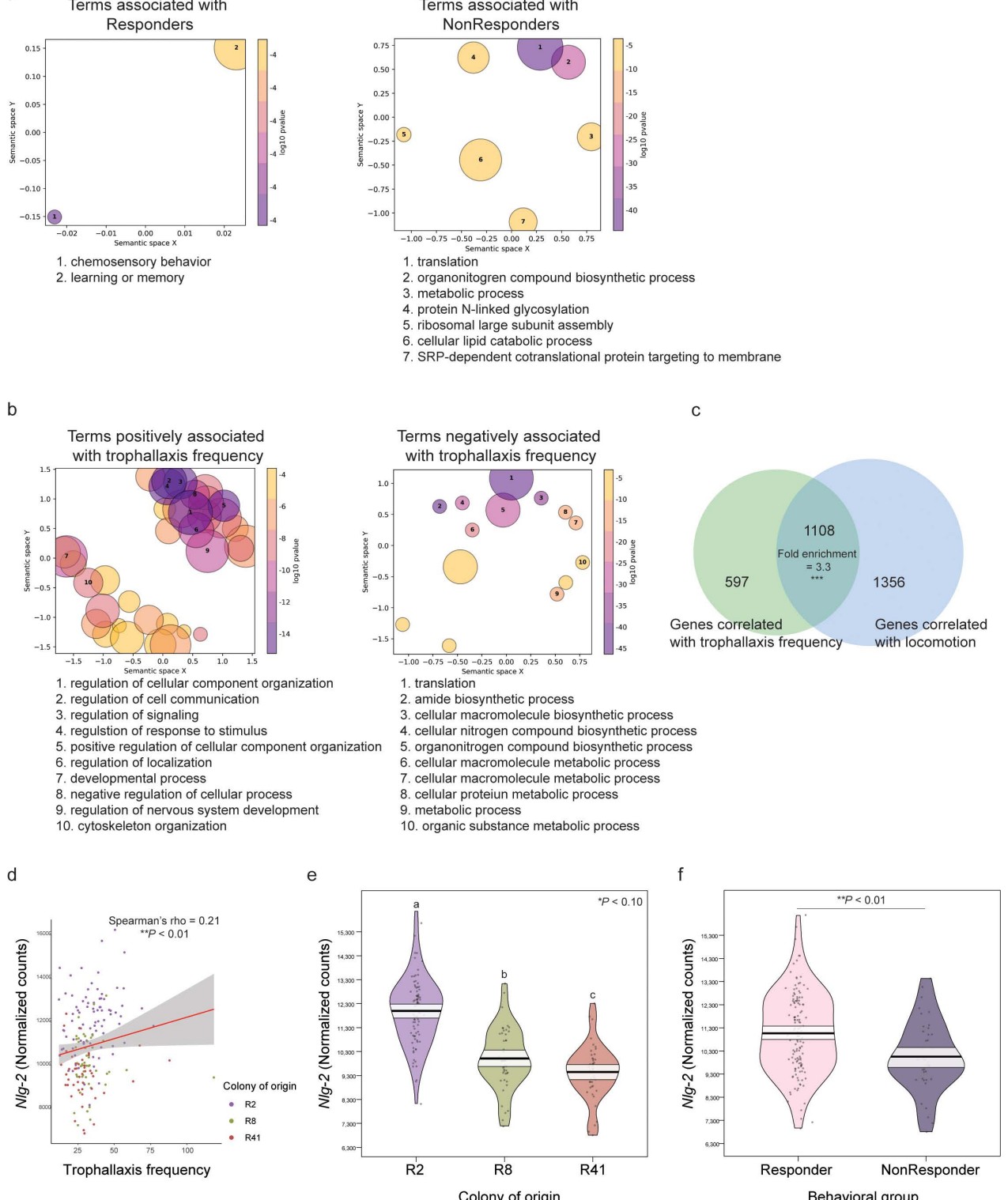

**Fig 3. Neuromolecular correlates of trophallaxis frequency and social responsiveness in the honey bee mushroom bodies (MBs). (a)** Gene Ontology (GO) analysis identified highly specific terms associated with chemosensation and learning enriched in genes upregulated in Responders, while metabolic process terms were enriched in genes upregulated in NonResponders. Note that for GO term circle coloration in **(a)**, all *P*-values are

within the same order of magnitude. **(b)** GO enrichment analysis performed on genes positively correlated with trophallaxis frequency revealed the enrichment of several terms related to cell signaling and stimulus response, while genes negatively associated with trophallaxis were more strongly related to biosynthesis and metabolism, and (c) this gene set was significantly similar to genes with expression levels correlated with locomotion (hypergeometric overlap test). **(d)** *Nlg2* levels were weakly but significantly correlated with trophallaxis frequency, (e) significantly lowest in colony R41 (one-way ANOVA with Tukey post-hoc test), and (f) differentially expressed in Responders compared to NonResponders in a lab-based assay examining behavioral responsiveness to social stimuli (Wilcoxon rank-sum test). For GO plots, circle diameter is inversely correlated to term specificity in the GO hierarchy, and more similar terms are more closely clustered in semantic space. Violin plots are constructed as previously described. Code and data underlying Fig 3 can be found in S2-S8 Tables and https://doi.org/10.6084/m9.figshare.29845490.

also identified a significant overlap of 1108 genes with expression values that correlated both with trophallaxis frequency and locomotion (hypergeometric test, fold enrichment = 3.3, *P* = 2.75e-239; Fig 3c and S8 Table). We also found that a major royal jelly protein (MRJP) precursor, *Mrjp6*, was significantly positively correlated with trophallaxis frequency but not locomotion, while *Mrjp 1*, *2*, *3*, *4*, *5, 7* and *8* were all negatively correlated with locomotion and no *Mrjps* were positively associated.

Comparing our data with previously published results, we did not identify a significant overlap of genes positively or negatively associated with trophallaxis frequency and those associated with food reward or anticipation of food reward (hypergeometric test, *P* > 0.10) [72,73].

### Neuroligin-2 *RNA-Seq expression and fluorescent* in situ *hybridization*

Motivated by the GWAS results, we explored MB *nlg2* expression and its relationship to other behavioral and genetic metrics. Expression levels were significantly associated with trophallaxis frequency (Spearman's rho = 0.21, *P* = 0.0063, Fig 3d), source colony (Tukey HSD post-hoc test on significant ANOVA [$F_{(2, 169)}$ = 69.762, *P* < 2e-16], *P* < 0.10 for each comparison, Fig 3e). Next, we compared *nlg2* expression in the MB in non-responders to the average expression in socially responsive bees, namely generalists, foragers, guards, and nurses, finding expression levels to be significantly higher in responders (Wilcoxon rank-sum test, W = 3,682, *P* = 0.0014, Fig 3f).

Following the association of *nlg2* and trophallaxis frequency in both GWAS (Fig 1b) and RNA-Seq results (Fig 3d-f), we asked where *nlg2* is expressed in the brain, how that localization changes over the course of development, and if its regional localization overlaps with regions known to be associated with social decision-making. We specifically compared 1-day-old "callow" workers, 1-week-old in-hive bees, and 2-week-old bees that are prepared to leave the hive to perform outdoor tasks. We used HCR-FISH to visualize *nlg2* expression. We found *nlg2* localization primarily in neuronal somata and largely absent from synaptically dense neuropil when compared to control samples where no gene-specific probes were used (Fig 4a and 4b), as expected. The Class I Kenyon cells of the MBs appeared to have higher expression, particularly in the densely packed neurons at the center of the MB calyces (Fig 4c), and expression was weaker in the surrounding, more loosely packed cells (Fig 4d). We also observed expression in MB extrinsic neurons and dorsal midline populations of cells that compose the pars intercerebralis (Fig 4e), a small population of large neurosecretory cells vital for nutritional signaling [74]. Some optic lobe cell populations, especially those most proximal to the central brain, also showed discernible fluorescence indicating transcription of *nlg2*.

As our study considered 10-day-old bees, we sought to qualitatively assess how spatiotemporal expression of *nlg2* varied before and after this age. We performed FISH on one-, seven-, and 14-day-old bees, and found that *nlg2* expression was present in 1-day-olds but weak in 7-day-old bees before returning to discernible levels in 14-day-old bees (Fig 4f-k).

## Discussion

Social proclivities vary within animal societies, driving a continuous distribution of individual tendencies to associate with others. This is true for many species, from humans to honey bees: some individuals pursue regular social contact and maintain large networks while others engage less. Social interactions are multifaceted, as are their drivers, which integrate

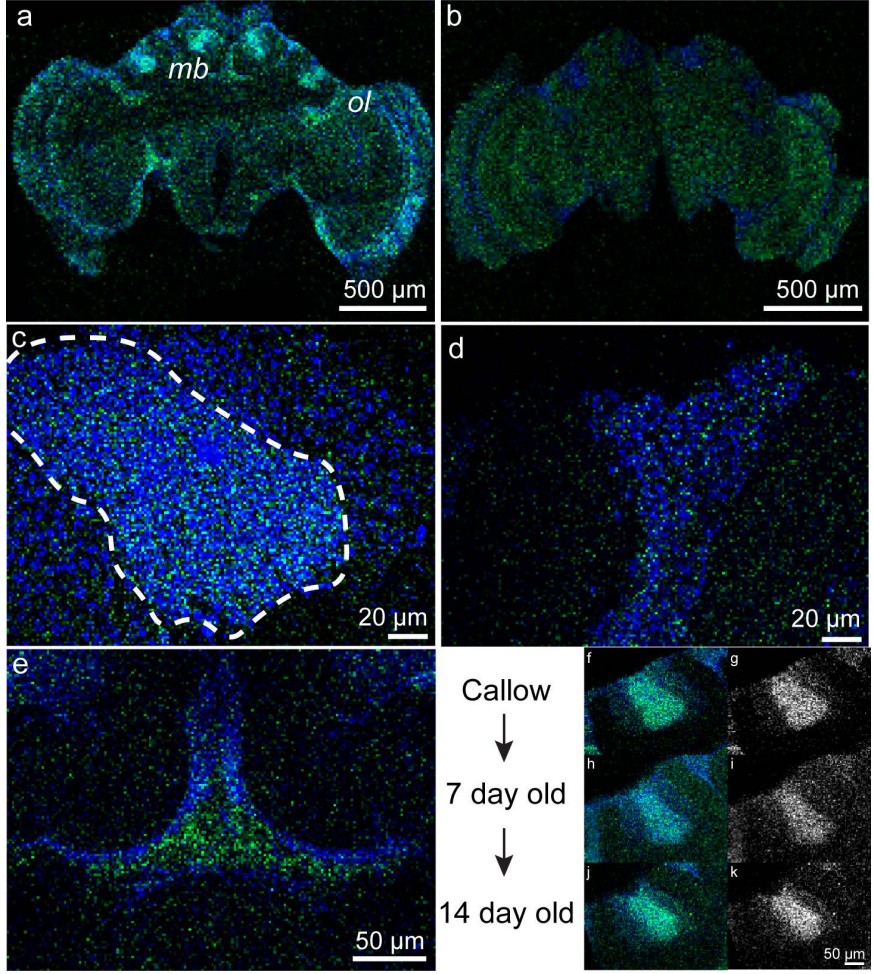

**Fig 4. Whole-mount fluorescent *in situ* hybridization reveals *neuroligin-2* (*nlg2*) expression patterns in the developing adult honey bee brain. (a)** *Nlg2* is widely expressed in the adult worker brain as detected via HCR probes, while **(b)** no staining was observed with antisense probes. **(c)** Expression levels were highest in the mushroom body (MB) Kenyon cells (KCs), specifically the inner-compact KC (dashed white line), yet **(d)** were observed in the outer-compact KC as well. **(e)** We also identified strong signaling along the dorsal midline of the brain, in or around the neurosecretory cells that compose the pars intercerebralis. **(f-k)** *Nlg2* expression was apparent throughout adulthood in the worker honey bee brain, although levels appeared highest in callows and 2-week-old bees (left and right columns are color and greyscale depictions of the same image, respectively). For each panel, *nlg2* localization is shown in green and nuclei counterstained with DAPI are shown in blue.

across physiology, developmental and motivational state, genetics, and past experience. It, thus, remains challenging to quantify social interactions and their molecular underpinnings. We used a combination of high-throughput behavioral tracking [31,42], laboratory-based behavioral assays [5,50], genome resequencing, brain transcriptomics, and comparative genomics to explore honey bee sociability and its molecular underpinnings. We focused on trophallaxis interactions and inter-individual motor coordination. Our results indicate deep homology in the molecular roots of social behavior shared between honey bees and vertebrates while also revealing divergent mechanisms.

Our GWAS identified 18 SNPs associated with variation in trophallaxis sociability. In contrast, no SNPs were identified when a GWAS was performed on locomotion or turning rate, kinematic metrics that contribute to trophallaxis. Like any social interaction, trophallaxis is more than the sum of its parts: rapid movement may support more social interactions but our results indicate it does not alone drive variation in tendency to interact in this way. By standards of human GWAS, our

sample size (n = 357 individuals) was small, but similar numbers of sociability-associated SNPs have been identified from GWAS with much larger human cohorts [29], reflecting the complexity of factors underpinning sociability.

Over half of the identified SNPs were localized to the intronic regions of genes. Using the Simons Foundation Autism Research Initiative (SFARI) gene database (https://gene.sfari.org/), we identified *neuroligin-2* (*nlg2*) and *NMDA receptor 2* (*nmdar2*) as Category 1 ("high confidence") genes, meaning they have been strongly implicated in human autism. Another gene with an intronic SNP, *nuclear factor 1 X-type* (*Nf1*) is characterized as a "Syndromic" autism gene, meaning its dysfunction is associated with the development of autistic traits that are not typically associated with a conventional autism diagnosis. In humans, impaired glutamatergic signaling by way of NMDAR disruption has clear associations with the development of autism [75], while glutamate is best known in insects for its role at the neuromuscular junction [76]. However, NMDAR activity is causally associated with memory formation in honey bees [77], and its function may, therefore, contribute to the reinforcement of trophallaxis relationships, which are not only likely shaped by inclusive fitness effects but may also be supported by reciprocity between nestmates.

The identification of two *nlg2* SNPs associated with variation in trophallaxis sociability contributes to a comparative framework for understanding the molecular building blocks of social behavior across highly diverged species. Neuroligins are cell surface proteins involved in the formation and remodeling of synapses in the central nervous system in an isoform-specific manner, and *nlg2* acts exclusively at inhibitory synapses [78]. In vertebrates, genomic variation in, or disrupted expression of, *nlg2* or its interaction partners is associated with forms of neurodivergence like autism, schizophrenia, and anxiety, each of which is linked to differences in how individuals associate with their social environment [2,79,80]. While we cannot assume the exact influence of these SNPs on phenotypic measures, including transcriptomics, intronic SNPs found in genes like *nlg2* may affect the stability, splicing, localization, or binding efficiency of regulatory elements, thus contributing to changes in gene expression and, thus, possibly behavior. Indeed, we report lower *nlg2* expression correlates with SNP prevalence at the colony level, indicating a potential mechanistic link between genetic sequence and molecular phenotype. We also suggest that, based on the findings of this study, trophallaxis frequency may serve as a measure of an individual's sociability and may be considered "trophallaxis sociability."

Several correlative findings suggest a role for *nlg2* in honey bee trophallaxis sociability. First, the two intronic variants (SNPs) in the honey bee *nlg2* gene that we identified were predominantly restricted to individuals from one source colony and those individuals displayed low levels of trophallaxis sociability compared to those from other source colonies. We probed other kinematic aspects of these bees and found that this lower level of sociability was not due to slower movement. Second, *nlg2* expression levels in the MBs, a higher-order sensory integration center in the bee brain [14,81], were also significantly correlated with trophallaxis rate and social responsiveness across all three tested colonies. Third, a strong signal of *nlg2* expression was localized to the MB and appeared to overlap the pars intercerebralis, a set of neurosecretory cells that mediate the relationship between diet, nutrition, and division of labor [74]. Trophallaxis sociability in honey bees illustrates the importance of food-sharing as a foundation for animal society. We speculate that *nlg2* may have played a role in the shaping of ancestral feeding circuitry to drive stable social network formation in this social bee lineage, as may be the case for other social animals. Consuming but not fully digesting food requires a neurobiological change to bypass the rewarding nature of feeding events, yet such a mechanism provides a foundation for the formation of long-term social bonds [82].

In fruit flies, *nlg2* is required for proper synaptic differentiation and function in the developing brain, and its dysregulation leads to aberrant neuronal signaling and impaired coordination of social behaviors [83–85]. Specifically, *nlg2*-deficient flies perform less courtship toward females, less aggression toward males, and keep greater distances from conspecifics. Like for bees, locomotor activity is unaffected, indicating distinct control mechanisms governing socially motivated behaviors and overall movement [85,86]. Beyond *nlg2*, considerable evidence points toward significant roles of neuroligins in the context of social recognition, stress reactivity and aggression in mice [87,88] and mate choice in swordtail fish (*Xiphophorus nigrensis*) [89,90], suggesting that a more detailed phylogeny of *neuroligin* gene family evolution across vertebrates

                                                                  

and invertebrates is needed to better understand its conservation and function in the context of social decision-making. For honey bees, we demonstrate that variants in this gene are associated with lower expression levels and less food-sharing overall, but we do not yet have a mechanistic understanding of how *nlg2* supports the observed behavioral variation. If *nlg2* coordinates the development and stability of inhibitory synapses in bees as it does in vertebrates, this indicates a significant role for inhibitory neurotransmitters like GABA, which has been hypothesized to regulate MB development during sensitive periods [91] in the tuning of social circuitry in the bee brain.

We also explored the synchronization of body movement between pairs of individual bees, some not necessarily engaged in trophallaxis. This model of social influence is similar to trophallaxis in that it not only requires coordination of one individual's body with another among honey bees located close to each other but also directly quantifies a measure of social influence identified in other species [70]. We found that individuals containing the *nlg2* allele associated with reduced trophallaxis also displayed reduced motor synchrony. This means that the movement of bees in the colony with high frequencies of this allele had less influence on, and was less influenced by, the movement of other bees. Inter-personal synchrony characterizes temporally or physically aligned behaviors in humans that naturally emerge between individuals in a social group, and such bidirectional motor influence tends to be diminished among autistic individuals and their peers [92,93]. It is interesting to note that casual observations revealed no motor deficits in the colony with high frequencies of this allele and, it is, therefore, unclear if negative fitness consequences exist. It may be that such variation is well tolerated within the honey bee population in the context of colony performance.

Leveraging genotypic information for each bee, we found that the genetic representation of variation in trophallaxis sociability also characterized variation in social influence. While we did not find a strong genotypic signature of social influence via GWAS, PRS derived from SNPs associated with trophallaxis sociability were negatively associated with an individual's propensity to influence or be influenced by nestmates. These results suggest a stronger role for immediate social environment in establishing this dimension of a honey bee's social phenotype, and genetic similarities between these two traits are also likely to exist.

Whole-mount FISH analysis identified predictably widespread distribution of *nlg2* in the worker brain. We also found that developmental changes in expression paralleled behavioral maturation, which itself is associated with variation in trophallaxis sociability [5,31]. Day-old workers have relatively high expression of *nlg2*, and these levels decrease in the first week of life as the bee performs in-hive tasks, rising again during the transition to out-of-hive activities like defense and foraging. Substantial MB growth occurs during this time [94,95], likely due to concurrent intrinsic and environmental drivers [96]. We speculate that temporal variation in signal intensity of *nlg2* corresponds to synaptic pruning mechanisms during the early phases of adult brain development, as neurodevelopmental patterning is shaped by intrinsic and external factors like developmental programming, motivated state, endocrine influences, and social environment. Moreover, we observed dense *nlg2* expression in the inner-compact Kenyon cells, a subregion of the MB previously found to be devoid of calcium/calmodulin-dependent protein kinase II (CaMKII) [97], a signaling molecule associated with synaptic plasticity, learning and memory expressed in much of the brain throughout development. It may be the case that *nlg2*-dependent synaptic formation relies more heavily on intrinsic, rather than learned, processes, the dysregulation of which contributes to variation in social behaviors.

Social interactions help establish and reinforce group membership and, in the context of food-sharing, facilitate mutually rewarding altruistic bonds between individuals [98,99]. In this regard, we predicted trophallaxis to be an intrinsically rewarding behavior, as is honey bee dance communication [72]. However, this prediction was not supported by comparative transcriptomic analysis. We did not identify gene expression similarities between the present study and previous MB transcriptomic characterization of reward processing [72,73]. Perhaps this finding is related to the fact that dopaminergic neurons, which contribute to reward seeking, expectation and acquisition in honey bees [100,101] are sparsely distributed in small clusters throughout the bee brain [102,103], earlier findings corroborated by more recent single-cell brain analyses [104,105]. Perhaps, the bulk sequencing of the MB performed in our study both diluted and excluded dopaminergic

neurons. It may also be that food donation versus reception, which we did not distinguish in this study, supports different brain gene expression signatures.

We detected strong transcriptomic similarity between non-responders from this study and soldier bees characterized in another study [71]. The honey bee soldier population constitutes a minority of the colony, individuals that are poised to rapidly attack incoming threats but do not consistently perform other activities in the beehive [106]. Non-responder bees [24] showed negligible interest in repeated presentation of affiliative or agonistic social stimuli despite no overt health or nutritional defects [5], and later were found to exhibit decreased frequency of trophallaxis interactions compared to nest-mates. The survival benefits offered to the honey bee colony by soldier bees illustrates a unique parallel, as a soldier's behavioral repertoire is narrow, but intensely performed, by necessity: a hyper-focused defensive force ready to engage a threat at any moment requires a proclivity toward threat responsiveness at the expense of other social engagements. Future studies should test the hypothesis that non-responders are better characterized as potential "hyper-responders" in the specific context of nest defense.

The physiology of trophallaxis is not fully understood, and variation in food-sharing among nestmates represents a con-stellation of molecular and physiological traits, including the status of exocrine tissues, such as the hypopharyngeal glands (HPGs) in the head that influence the nutritional and signaling content of liquid food regurgitate [107]. In this study, the degree of HPG development was not investigated due to technical limitations of accessing the MB for gene expression, and so the possibility that HPG activity influences trophallaxis frequency cannot be ruled out. For example, we observed consistent differences in the relationship between MRJP gene expression and behavior. The strong, significant overlap of gene expression associated with trophallaxis frequency and locomotion suggests similar molecular mechanisms drive physical activity, as expected, yet one MRJP was strongly associated with trophallaxis frequency while seven others were negatively associated with locomotion. Although being highly pleiotropic, MRJPs likely affect behavior by influencing nutri-tional and hormonal status [108], and our results indicate a potential role for MRJPs in modulating the interplay between overall physical activity and the generation of trophallaxis [20].

While we identified significant variation in social proclivities across and within honey bee colonies and have started to elucidate its molecular roots, the effects of this variation on colony health and well-being are unknown. Basic evolu-tionary theory supports the speculation that socially divergent phenotypes may persist if the cost of purging maladaptive aspects of the phenotype outweigh benefits to the individual or group, the latter more strongly applying to the honey bee society. It may also be the case that such phenotypes are either tolerated or preserved across diverse animal lineages due to supportive, or perhaps even vital, roles they contribute to their society. It could be useful to explore this issue in honey bees, given serious concerns about honey bee health and worldwide colony population declines over the past two decades.

The expression of individual differences in social behavior, from humans to honeybees, is derived from a constella-tion of interacting neurobiological and physiological factors. While no single gene or genetic variant alone drives socia-bility, the identification of shared genomic underpinnings of social interactivity across diverse taxa hints at conserved molecular building blocks of social life. We anticipate new tools for automated tracking will continue to improve the granularity of such behavioral comparisons across social animals [32,42], further promoting a role for social insects in elucidating how the brain and genome have evolved to support the intricate social lives observed across diverse animal societies.

## Supporting information

**S1 Fig. Zygosity for trophallaxis frequency-associated single nucleotide polymorphisms (SNPs).** All SNPs are described in S1 Table; 17 out of 18 SNPs are shown, with the remaining SNP shown in Fig 2a. Gene identifiers are listed for intronic SNPs. Code and data underlying S1 Fig can be found at https://doi.org/10.6084/m9.figshare.29845490. (TIF)

**S2 Fig.  Genotype-specific variation in motor synchrony. information partial directed coherence (iPDC) was applied to pairs of bees' *X* and *Y* coordinate time-series, obtaining information flow (*I*flow) between the movement of a bee and its neighbor along the X and Y axis, respectively.** *I*flow was then averaged across the two coordinates obtaining a single *I*flow for each bee (Fig 2h). Violin plots are constructed as follows: points represent raw data, solid black lines represent the mean, pale white shading above and below the mean represent a 95% confidence interval, and plot shape represents a smoothed density curve outlining the distribution of raw data. Letters above violin plots represent significance ($P$-value < 0.05) from a between-group Tukey post-hoc analysis following a one-way ANOVA (for $X$ coordinate $I_{flow\ bee->neighbor}$: $F_{(2,352)}$ = 8.03, $P$ = 3.87e-04, $I_{flow\ neighbor->bee}$: $F_{(2,352)}$ = 7.02, $P$ = 0.001); for $Y$ coordinate $I_{flow\ bee->neighbor}$: $F_{(2,352)}$ = 6.92, $P$ = 0.001, $I_{flow\ neighbor->bee}$: $F_{(2,352)}$ = 1.4, $P$ = 0.248). Code and data underlying S2 Fig can be found at https://doi.org/10.6084/m9.figshare.29845490.
(TIF)

**S3 Fig.  (a) Principal components analysis of top 500 most varied genes across all samples; top left shows PCA for all three source colonies (R2, R8, and R41) remaining plots represent PCAs for each individual colony.** Each letter represents an individual bee assigned to a specific behavioral group based on automated observations of foraging and manual observations of behavior in a lab-based dish assay, as described in Methods. Behavioral groups are denoted as follows: Ge, Generalist; F, Forager; N, Nurse; G, Guard; NR, NonResponder. (b) Gene list overlap heatmap comparing mushroom body transcriptomic profiles of Generalists, Foragers, Nurses, Guards, and NonResponders in the present study to whole-brain transcriptomic profiles of Soldiers and Foragers from Traniello and colleagues (2023). Odds ratio is calculated via hypergeometric overlap test and resulting $P$-values are reported following correction with the Bonferroni method; N.S., non-significant. Code and data underlying S3 Fig can be found in S2-S8 Tables and https://doi.org/10.6084/m9.figshare.29845490.
(TIF)

**S1 Table.  Annotations and test statistics for 18 single nucleotide polymorphisms (SNPs) associated with variation in trophallaxis frequency via genome-wide association study (GWAS).**
(XLSX)

**S2 Table.  Differentially expressed genes detected in non-responders relative to remaining behavioral groups.**
(CSV)

**S3 Table.  Differentially expressed genes detected in generalists relative to remaining behavioral groups.**
(CSV)

**S4 Table.  Differentially expressed genes detected in foragers relative to remaining behavioral groups.**
(CSV)

**S5 Table.  Differentially expressed genes detected in guards relative to remaining behavioral groups.**
(CSV)

**S6 Table.   Differentially expressed genes detected in nurses relative to remaining behavioral groups.**
(CSV)

**S7 Table.  Genes with expression levels that correlate with trophallaxis frequency.**
(CSV)

**S8 Table.  Genes with expression levels that correlate with locomotion.**
(CSV)

## Acknowledgments

The authors thank A. Hernandez and C. Wright at the Roy J. Carver Biotechnology Center (UIUC) for sequencing services and members of the Robinson and Cook labs for valuable comments and feedback which improved the manuscript.

## Author contributions

**Conceptualization:** Ian Michael Traniello, Gene E Robinson.

**Data curation:** Ian Michael Traniello, Arian Avalos, Michael J. M. Gachomba, Tim Gernat, Adam R Hamilton.

**Formal analysis:** Ian Michael Traniello, Arian Avalos, Michael J. M. Gachomba.

**Funding acquisition:** Jennifer L Cook, Gene E Robinson.

**Investigation:** Ian Michael Traniello, Arian Avalos, Zhenqing Chen, Amy C Cash-Ahmed.

**Methodology:** Ian Michael Traniello, Michael J. M. Gachomba.

**Project administration:** Jennifer L Cook.

**Supervision:** Amy C Cash-Ahmed, Jennifer L Cook, Gene E Robinson.

**Visualization:** Ian Michael Traniello.

**Writing—original draft:** Ian Michael Traniello, Jennifer L Cook, Gene E Robinson.

**Writing—review & editing:** Ian Michael Traniello, Arian Avalos, Tim Gernat, Zhenqing Chen, Jennifer L Cook, Gene E Robinson.

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
