## [Editor Report · Decision Letter 0]

21 Feb 2025

Dear Dr Traniello,

Thank you for submitting your manuscript entitled "Molecular analyses of individual variation in honey bee sociability" for consideration as a Research Article by PLOS Biology.

Your manuscript has now been evaluated by the PLOS Biology editorial staff, as well as by an academic editor with relevant expertise, and I'm writing to let you know that we would like to send your submission out for external peer review.

IMPORTANT: We think that your study would be best considered as a Short Report. No reformatting is required, but please select "Short Reports" as the article type when you upload your additional metadata (see next paragraph).

Once your full submission is complete, your paper will undergo a series of checks in preparation for peer review. After your manuscript has passed the checks it will be sent out for review. To provide the metadata for your submission, please Login to Editorial Manager (https://www.editorialmanager.com/pbiology) within two working days, i.e. by Feb 25 2025 11:59PM.

Kind regards,

Roli Roberts

Roland Roberts, PhD

Senior Editor

PLOS Biology

rroberts@plos.org

---

## [Decision Letter · Decision Letter 1]

9 Apr 2025

Dear Ian,

Thank you for your patience while your manuscript "Molecular analyses of individual variation in honey bee sociability" was peer-reviewed at PLOS Biology. It has now been evaluated by the PLOS Biology editors, an Academic Editor with relevant expertise, and by four independent reviewers.

In light of the reviews, which you will find at the end of this email, we would like to invite you to revise the work to thoroughly address the reviewers' reports.

As you will see below, reviewer #1 liked the paper, but feels that its components are “disjointed” and that the framing needs work. There are several instances where s/he feels that additional analyses using available data (e.g. GWAS on individual-level trophalaxis data, controlling for task performance…) would improve the paper. Reviewer #2 is positive and has a modest number of requests, several regarding neuroligin orthology. Reviewer #3 is broadly positive, but makes a similar point to reviewer #1 regarding sociability vs task performance, and also asks about physiological state (royal jelly). S/he suggests several analyses, some of which may be possible. Reviewer #4 is very positive and has only textual requests.

Given the extent of revision needed, we cannot make a decision about publication until we have seen the revised manuscript and your response to the reviewers' comments. Your revised manuscript is likely to be sent for further evaluation by all or a subset of the reviewers.

**IMPORTANT - SUBMITTING YOUR REVISION**

*Re-submission Checklist*

*Published Peer Review*

*PLOS Data Policy*

*Blot and Gel Data Policy*

Sincerely,

Roli

Roland Roberts, PhD

Senior Editor

PLOS Biology

rroberts@plos.org

REVIEWERS' COMMENTS:

Reviewer #1:

[see also the fully formatted version in the attachment]

I enjoyed reading this manuscript and found the results exciting. At the same time, the different subsections of the results felt disjointed and I was confused by aspects of the overall framing.

The authors use a GWAS to show that 18 SNPs correlate with the frequency at which honeybees engage in trophallaxis. Two of these SNPs are in the gene neurolignin-2 (nlg-2), which is a focus of the paper.

In a following section, the authors show that bees that trophallax more frequently have a stronger influence on the movement of neighboring bees ('social influence') than bees that trophallax less frequently. This data is presented at the colony-level, not the individual-level even though the individual-level data exists. There could be an equivalent GWAS for these individual-level data, as was done for trophallaxis. My sense is that it might even be more informative, for reasons that are provided below. In the current form this results subsection feels disconnected.

The authors next look at gene expression in the mushroom body and find that 1705 genes correlate with trophallaxis frequency. It is not stated whether nlg-2 is among these. Instead, a separate analysis is performed to test whether nlg-2 expression correlates with trophallaxis frequency and behavioural role in the colony. The data on behavioral role is based on a suite of assays and measures, the data from which seems to be used only here, and even here only in passing, which is a shame because they are interesting.

In a last results section, the authors use FISH to localize the expression of nlg-2 to neuronal populations that are dispersed throughout the brain and show that expression is high in bees that are one-day old, negligible in bees that are one-week old, and elevated again in bees that are two-weeks old. This pattern is odd but is not discussed further. It is also not clear why the authors focus on age here, rather than sociability. It is the first time age is discussed.

My concerns are:

1) The authors use trophallaxis as a measure of 'sociability'. They control for movement kinetics, but not task behavior. I believe trophallaxis frequency varies with behavioural role (this is even written on L. 85). Do nurses and foragers trophallax at the same rate? I know there is evidence in ants that certain individuals specialize in food dissemination. I do not think these are more 'social'. It would be important to see how trophallaxis frequency relates to the performance of different tasks. The authors have the data. This concern would also be allayed if the authors have data showing that trophallaxis frequency does not vary with age. Related to this point:

- L.82: "In addition to task-related behaviors, honey bees frequently engage in mouth-to-mouth sharing of liquid…" I would disagree with this framing, and rather see trophallaxis as task-related.

- L. 90: "Taken together, trophallaxis is a quantifiable analog to other measures of sociability across diverse taxa". I would like to see more evidence of how trophallaxis varies for individuals over time, in association with task performance, and with other measures of sociability to justify this claim.

- L. 107: "we quantified movement kinetics to disentangle trophallaxis engagement frequency from overall activity rate". This is a great control, but the missing control of task performance is an issue.

2) The connection between the results subsections generally seem weak. The social influence analysis is tied in tangentially, and at a colony level while the trophallaxis analysis is at the individual level. The analysis of task behavior is almost not used, and the FISH data quantifies age-associated differences rather than differences in task behavior or trophallaxis frequency.

3) In the discussion there are a few points where the authors make evolutionary claims that I believe are unfounded:

- L. 500: Is there evidence for the idea that 'trophallaxis relationships rely on reciprocity between nestmates'? I expect the behavioural dynamics of social insect societies are shaped entirely by inclusive fitness effects, and not at all by reciprocity.

- L. 550: "Basic evolutionary theory supports the speculation that socially divergent phenotypes are either tolerated or preserved across diverse animal lineages due to supportive, or perhaps even vital, roles they contribute to their society". I do not believe that basic evolutionary theory supports that notion! It does not contradict it either, but there are a whole host of possible explanations for the presence of socially divergent phenotypes. For example, socially divergent phenotypes could be maladaptive but persist if the rate of developmental error is low and the cost of reducing the error rate further is high. Which is likely the reality.

Some minor additional points:

L. 80: "tracking now allows for behavioural monitoring… over long portions of [social insects] lives": There are ant papers which do exactly that, but the cited example is not one of them.

L. 151: Trophallaxis events are identified with a CNN. It would be useful to quantify the accuracy of this by comparing with manual observations.

L. 226: The automated detection of foragers is great. And the other behavioural assays. This data should be controlled for when performing GWAS on trophallaxis frequency.

L. 433: "Non-responders were transcriptomically distinct from generalists, foragers, guards, and nurses". To assess this, it would be useful to see the RNA-Seq data plotted out on a PCA with individuals from these different groups highlighted.

L. 467: This temporal pattern is strange. Does it relate to behaviors that old and young bees share and that middle-aged bees do not exhibit?

L. 453-454: Part of this sentence is missing: "when ("responders," Wilcoxon…"

L. 450: The link between this sentence and the next is confusing. The first states that the authors are testing whether nlg expression relates to task performance. The next sentence states that expression levels were correlated with 'trophallaxis sociability', without providing the result from the test that the first sentence stated was being done!

L. 508: It would be useful to have a more detailed explanation of which specific Drosophila social behaviors are affected by mutations in nlg-2.

Reviewer #2:

In the manuscript entitled "Molecular analyses of individual variation in honey bee sociability," Traniello et al. combine genome sequencing, brain transcriptomics, and automated behavioral tracking to identify 18 SNPs associated with variation in trophallaxis (which they consider a proxy for "sociability") in honey bee colonies. Several of these SNPs were associated with genes previously linked to sociability in other species. The authors also find genes whose expression covaried with sociability. Overall, this is a timely and interesting study. The manuscript is well written. I have only a few comments which are intended to help he authors to improve the manuscript further.

General comments:

1) The concordant discovery of neuroligin-2 (nlg2) in both genomic screen and transcriptome analysis is obviously cool. In the absence of a detailed and careful phylogenetic analysis of the entire neuroligin gene family I am wondering how confident we can be about the orthology assignments with putative orthologs in Drosophila and humans? I think such an analysis is beyond the scope of the current study, but I certainly hope someone will do this analysis at some point. Interestingly enough, neuroligins have also been implicated in female mate choice across teleost fishes (see, for example, https://doi.org/10.1242/jeb.207324,
https://doi.org/10.1111/j.1601-183X.2011.00742.x,
https://doi.org/10.1159/000360071,
https://doi.org/10.1098/rspb.2024.0121).

2) Did the authors look at other "sociability" candidate (from work in bees, Drosophila, and/or vertebrates) genes in their transcriptomic data? It would be interesting to see whether other candidates show intriguing patterns similar to neuroligin.

3) In the Discussion, can the authors speculate what neuroligin might be doing in honey bees to drive aspects of social behavior?

Specific comments:

- Line 122: Should " instrumentally" be replaced with "artificially" or similar?

- Lines 130ff: Sample sizes need to be clearly stated.

- Lines 149ff: This automated detection of trophallaxis events is cool, but I am wondering whether this has been compared to manually scored behavior for some of the videos. In other words, how do the authors know that trophallaxis events involve two bees 1.7 mm - 7.4 mm apart and lasting between <3 seconds or >3 seconds?

- Line 306: 13,000 paired-end reads/sample seems rather low? How many genes?

- Lines 316ff: How many genes were analyzed? What filter criteria (i.e., above which TPM threshold in what portion of samples) were applied? How was the data normalized?

- Line 367: How was "robust one-to-one orthology" confirmed? I assume by reciprocal BLAST? In Drosophila, both nlg-2 and nlg-3 have been implicated in aspects of social behavior. While beyond the scope of the current study, long-term, it will be good to assemble a phylogeny for this gene family.

- Figure 5: The micrographs are very pixelated, which makes it difficult to assess these results.

Reviewer #3:

In this manuscript, the authors used a combination of automated behavioral tracking, GWAS, brain transcriptomic analyses, and FISH analysis to suggest that the trophallaxis frequency in honey bees correlates with mutations and expression levels of Nlg2, which is related to human sociability, suggesting the possibility that there is a common molecular neural basis for honey bee and human sociability. The notion that trophallaxis could be an indicator of honey bee "sociability" is novel, and the large part of the experiments aimed at identifying GVAS and gene expression variation correlated with trophallaxis frequency are well designed. On the other hand, the experimental evidence supporting the notion that trophallaxis frequency could be an indicator of honey bee sociability seems still not so much robust and needs further substantiation. Therefore, the possibility that trophallaxis frequency may correlate with traits other than "sociability" needs be tested in this manuscript as well, which will make the conclusions of this paper more objective and convincing. My comments are as follows.

Major comments:

1. Besides "sociability", the trophallaxis frequency in worker honey bees may also reflect the physiological status of individual worker, as stated in the Introduction section (Line 35). Since the fluid exchanged by trophallaxis in nurse bees is royal jelly secreted from the head glands (Crailsheim 1998), trophallaxis frequency may reflect the individual differences in the amount of royal jelly produced in nurse bees. (If some workers have low royal jelly production due to undeveloped head glands, would they still perform frequent trophallaxis?) To test this possibility, the authors need to test if there is any correlation between the development of the head glands (or the expression levels of royal jelly protein gene) with trophallaxis frequency (if the authors' notion is correct, we would expect no significant correlation). Alternatively, if the tendency of some workers to show high or low trophallaxis frequency does not change throughout their life (from nurse nees to foragers), this may also suggest that trophallaxis frequency does not mainly reflect a physiological state but "sociability (by nervous system)" of the bees.

If it is difficult to perform this experiment, then the Discussion should clearly state that this alternative possibility is not tested, and the entire manuscript needs to be rewritten to take into account the possible contribution of individual differences in physiological status as well as "sociability (by the nervous system)" to the trophallaxis frequency. Currently, the manuscript is written on the premise that trophallaxis frequency is an indicator of "sociability" and needs to be toned down significantly.

2. For Figs. 2 and 3; when the authors obtained the results that individuals from colony R41 are homozygous for the Nlg2 allele, while individuals from colonies R2 and R8 are predominantly heterozygous, they should have performed a similar analysis for wild type individuals derived from colonies R2, R8 or other colonies. I assume that the authors have this data, so I would like to strongly suggest the authors to reanalyze the data and include the results in this manuscript.

Minor comments:

1. For the reasons stated in my Major comment 1, the term "trophallaxis sociability" should be replaced with a term "trophallaxis frequency" for the entire manuscript.

2. Please consider showing the mutations in the intron of the Nlg2 gene in a supplementary Figure and discuss the possible mechanism by which the mutations alter the expression levels of Nlg2.

3. In FISH experiments, Nlg2 is also strongly expressed in inner-compact KCs and optic lobes. Please describe the presumed function of Nlg2 in these neurons.

4. The values on the left vertical axis in Fig. 4a are all -4?

Reviewer #4:

The interesting manuscript by Traniello et al. investigates individual differences in honey bee social behavior and their underlying molecular correlates. The authors identify genetic and molecular correlates underlying trophallaxis in honey bees, a highly social insect model by combining genome sequencing and brain transcriptomics with behavioral tracking. Honey bees which were "socially unresponsive" had shorter and briefer trophallaxis interactions compared to age-matched individuals which were more responsive to social stimuli in an earlier study. The authors of the current study reveal 18 SNPs which are associated with some variation in social behavior. Some of these are localized to genes associated with social behavior in other species and the authors suggest fundamental and shared molecular mechanisms underlying sociability. The identification of two nlg2 SNPs associated with variation in trophallaxis is particularly exciting, because these genes have been associated with human autism. The link between nlg2, reduced motor synchrony between individual honey bees and human autism appears less convincing, though. Overall, this study reveals exciting new insight into the molecular mechanisms underlying social behavior, although, naturally, honey bees are evolutionarily very distant from humans.

The manuscript is very well written. I only have a couple of minor points to consider.

Minors

1)Line 46

There are other factors impacting task performance in honey bees, please add physiology and hormones here and add appropriate citations (maybe including some outside the Robinson group)

2) Line 146-153

What was the error date in the detection of bar-coded bees? How well can bees be distinguished if they are on top of each other?

3)Lines 252

Why did you use the thorax for DNA sequencing and not the brain?

Did you pool samples for DNA sequencing?

4) Line 257ff

Did you pool mushroom bodies for RNAseq? If so, how did you pool?

5) Why did you only study 13,000 reads of brain MBs? Is there a writing mistake and you mean 13,000,000? It is still low. Why did you not sequence deeper with a small amount of tissue and a comparatively low gene expression?

---

## [Decision Letter · Decision Letter 2]

1 Aug 2025

Dear Ian,

Thank you for your patience while we considered your revised manuscript "Molecular analyses of individual variation in honey bee sociability" for publication as a Short Report at PLOS Biology. This revised version of your manuscript has been evaluated by the PLOS Biology editors, the Academic Editor, and two of the original reviewers.

Based on the reviews, we are likely to accept this manuscript for publication, provided you satisfactorily address the remaining points raised by the reviewers and the following data and other policy-related requests.

IMPORTANT - please attend to the following:

a) Please change your Title to something more explicit and declarative. We suggest something like "Genetic variation influences food-sharing-based sociability in honey bees" (happy to discuss by email).

b) Please attend to the remaining requests from reviewer #3.

c) Sorry to raise this at such a late stage (my bad!), but the Short Report format only allows four Figures. You currently have five, so perhaps you could combine two of them (Figs 1 and 2 are relatively simple in structure, so would seem good candidates for this?).

d) Please include the URLs for your funder(s) in the Financial Disclosure section of the submission form.

e) Please address my Data Policy requests below; specifically, we need you to supply the numerical values underlying Figs 2, 3ABCGHI, 4ABCDEF, S1, S2, S3AB, either as a supplementary data file or as a permanent DOI’d deposition.

f) Please cite the location of the data clearly in all relevant main and supplementary Figure legends, e.g. “The data underlying this Figure can be found in S1 Data” or “The data underlying this Figure can be found in "https://zenodo.org/records/XXXXXXXX"

g) Please make any custom code available, either as a supplementary file or as part of your data deposition.

We expect to receive your revised manuscript within two weeks.

*Published Peer Review History*

*Press*

Sincerely,

Roli

Roland Roberts, PhD

Senior Editor

rroberts@plos.org

PLOS Biology

DATA POLICY:

Regardless of the method selected, please ensure that you provide the individual numerical values that underlie the summary data displayed in the following figure panels as they are essential for readers to assess your analysis and to reproduce it: Figs 2, 3ABCGHI, 4ABCDEF, S1, S2, S3AB. NOTE: the numerical data provided should include all replicates AND the way in which the plotted mean and errors were derived (it should not present only the mean/average values).

CODE POLICY

DATA NOT SHOWN?

REVIEWERS' COMMENTS:

Reviewer #1:

[identifies himself as Tomas Kay]

I am greatly impressed by the changes that the authors have made in response to the first round of reviews. They have done an excellent job. All concerns have been thoroughly and thoughtfully addressed and I am excited to see this manuscript published.

Reviewer #3:

I appreciate that the authors took my comments seriously and attempted to improve the manuscript. However, I still find several points that I believe need further revision. My comments on the authors' responses to my previous comments are as follows.

Major comment 1:

I understand that the previous samples are unavailable. That is unavoidable. However, in that case, it is necessary to state in the Discussion section that "In this study, the degree of development of the hypopharyngeal glands in the samples was not investigated, and so the possibility that the degree of hypopharyngeal gland development influences trophallaxis frequency cannot be ruled out."

In the revised manuscript, lines 535-537 and 700-711 have been added to include the information that "the mushroom body gene list includes major royal jelly protein 6 precursor." However, while this may suggest a new function of major royal jelly protein 6 precursor in the mushroom body, it does not indicate the degree of development of the hypopharyngeal glands in the samples (at least, there is no evidence to support this). Therefore, while it is acceptable to add lines 535-537 and 700-711, it is also necessary to include the following statement in the Discussion section: "In this study, the degree of development of the hypopharyngeal glands in the samples was not investigated, and so the possibility that the degree of hypopharyngeal gland development influences trophallaxis frequency cannot be ruled out."

Major comment 2:

I understand the authors' explanation.

Minor comment 1:

I understand that the authors wish to use the term "trophallaxis sociability" in important sections to emphasize the main theme of the paper. However, I do not agree that this term should be used in the Introduction or Results section, as it should be used in the Discussion of this paper, where the results of this paper indicate that the Nlg2 gene is associated with trophallaxis frequency. While I appreciate that the authors have removed most instances of the term "trophallaxis frequency" from the Introduction and Results sections, two instances remain in the Introduction section and have not been removed from the figure legends. I believe it is necessary to remove these terms "trophallaxis frequency" from these portions first.

On the other hand, for example, after adding the following sentence at the end of the fourth paragraph of the Discussion (line 605): "Based on the findings of this study, trophallaxis frequency may serve as a measure of an individual's sociability, and in that sense, trophallaxis frequency could potentially be rephrased as 'trophallaxis sociability.'" I believe it would be appropriate to use the term "trophallaxis sociability" in the subsequent sections of the Discussion.

Minor Comment 2:

The author misunderstood my request due to my poor explanation. I simply requested to explain the possible mechanisms by which mutations in the intron of Nlg2 could affect the transcription level of Nlg2. The description in lines 609-612 does not address my request, so please reconsider. Has the mutant sequence of the Nlg2 gene already been shown in the data?

Minor comment 3:

I understand the authors' explanation.

Minor Comment 4:

I understand the authors' explanation.

---

## [Editor Report · Decision Letter 3]

13 Aug 2025

Dear Ian,

Thank you for the submission of your revised Short Report "Genetic variation influences food-sharing sociability in honey bees" for publication in PLOS Biology. On behalf of my colleagues and the Academic Editor, Lars Chittka, I'm pleased to say that we can in principle accept your manuscript for publication, provided you address any remaining formatting and reporting issues. These will be detailed in an email you should receive within 2-3 business days from our colleagues in the journal operations team; no action is required from you until then. Please note that we will not be able to formally accept your manuscript and schedule it for publication until you have completed any requested changes.

Sincerely, 

Roli

Senior Editor

PLOS Biology

rroberts@plos.org